# Center-Based Approximation of a Drifting Distribution

**Alessio Mazzetto**                                    ALESSIO_MAZZETTO@BROWN.EDU
*Brown University*

**Matteo Ceccarello**                                   MATTEO.CECCARELLO@UNIPD.IT
**Andrea Pietracaprina**                             ANDREA.PIETRACAPRINA@UNIPD.IT
**Geppino Pucci**                                             GEPPINO.PUCCI@UNIPD.IT
*University of Padova*

**Eliezer Upfal**                                          ELIEZER_UPFAL@BROWN.EDU
*Brown University*

**Editors:** Gautam Kamath and Po-Ling Loh

## Abstract

We present a novel technique for computing a center-based approximation of a drifting distribution. Given $k \geq 1$ and a stream of data, whose distribution is changing over time, the goal is to compute, at each step, the best $k$ centers representation of the current distribution, despite possibly having only a single sample from the most recent distribution. In data mining, this is traditionally attempted through the sliding-window mechanism, where the analysis is performed on the most recent fixed-size segment of the data. The problems with this approach are twofold: (1) setting the correct window size is challenging; and (2) a fixed window size cannot effectively track changes in the distribution happening at variable speed. In this paper, we propose a new methodology that dynamically adjusts the window size based on the recent drift of the data. The challenge is that it is not possible to explicitly estimate the drift, as we may have only a single data point from each distribution. Our main contribution lies in providing a rigorous mathematical analysis, establishing both an upper bound via a dynamic window size algorithm, and a lower bound that shows the tightness of our approach.

## 1. Introduction

We study the problem of approximating a drifting distribution using $k$ points. Given a sequence of independent random vectors $(X_t)_{t \in \mathbb{N}}$ from $\mathbb{R}^d$, where $X_t$ is a single sample from the distribution $P_t$, the goal is to compute, at each step $T$, a set of $k \geq 1$ centers that best represent the current distribution $P_T$ using only the samples $X_1, \ldots, X_T$. This problem has been extensively studied for a fixed distribution $P$. Given an integer $k \geq 1$ and a real $z \geq 1$, the objective is to represent $P$ using only $k$ points, dubbed *centers*, such that the expectation of the $z$-th power distance between a point drawn from $P$ to its closest center is minimized. Formally, the goal is to solve (or approximate) the following minimization problem:

$$\min_{\substack{C \subset \mathbb{R}^d: \\ |C|=k}} \mathcal{W}_{k,z}(C,P) \text{ with } \mathcal{W}_{k,z}(C,P) = \int_{x \in \mathcal{X}} \min_{c \in C} \|x - c\|^z dP(x) \ , \tag{1}$$

where $\mathcal{W}_{k,z}(C,P)$ can be regarded as the cost of approximating $P$ using the set of centers $C$. As an important special case, the optimal center for $k = 1$ and $z = 2$ is the mean of the distribution $P$. Throughout this paper, we refer to the optimization problem (1) as the $(k,z)$-*approximation problem* for a distribution $P$.

The problem has been studied (mostly for $z = 2$) under the name of *vector quantization* (Bartlett et al., 1998; Linder, 2000, 2002; Levrard, 2013, 2015), or *clustering* of a distribution (Biau et al., 2008; Maurer and Pontil, 2010; Maurer, 2016; Foster and Rakhlin, 2019; Li and Liu, 2021; Bucarelli et al., 2023). Indeed, for a fixed distribution $P$, the $(k, z)$-approximation problem is related to the popular problem of $(k, z)$-clustering of pointsets (Gupta and Tangwongsan, 2008; Huang et al., 2018; Borassi et al., 2020; Cohen-Addad et al., 2021b). Specifically, a $(k, z)$-approximation of a fixed distribution $P$ can be obtained by $(k, z)$-clustering a sufficiently large sample from $P$ (Biau et al., 2008; Liu, 2021). Vice versa, the $(k, z)$-clustering of a finite pointset $S \subseteq \mathbb{R}^d$ is equivalent to solving the $(k, z)$-approximation problem for the uniform distribution over $S$. Note that for $z = 1$ and $z = 2$, this amounts to computing a solution to the well known $k$-median and $k$-means problems, respectively.

In this work, we study the $(k, z)$-approximation problem in the context of streaming data with concept drift that evolves over time. Specifically, the goal is to solve the $(k, z)$-approximation problem for the distribution $P_T$ at time $T$ based on the input points $\{X_1, \ldots, X_T\}$ observed so far, which have been sampled independently from a sequence of unknown distributions $P_1, \ldots, P_T$. Since we have only a single sample $X_T$ from $P_T$, a solution to this problem must rely on previous samples. However, unlike the static case, where the distribution remains unchanged, using a $(k, z)$-clustering solution over *all* the past samples $\{X_1, \ldots, X_T\}$ from a drifting distribution can result in a poor approximation of the distribution $P_T$. In fact, since the distribution of the data can change over time, it may be the case that only the most recent data is relevant, and the old data should not influence the current solution. In the learning community, this phenomenon is referred to as *concept drift*, which has been identified as "*a major problem in online learning caused when a model based on old data fails to correctly reflect the current state of the world*" (Babcock et al., 2003).

Several approaches have been proposed to address concept drift, all sharing the underlying principle of assigning greater relevance to more recent points, typically by applying a decaying weight to older data. A particularly popular framework in this context is the *sliding window model* (Braverman, 2016), where at any given time, a solution is computed solely based on the most recent $w$ points, where $w$ is an integer representing a fixed *window size*. The parameter $w$ plays a crucial role in determining the quality of the solution. Intuitively, a large $w$ may include outdated points due to concept drift. Conversely, if $w$ is too small, the solution may be based on an insufficient number of points, leading to poor generalization. Ideally, the window size should be large enough to contain sufficient data for robust analysis, yet small enough to exclude points that are no longer relevant. This optimal window size may vary over time, particularly in cases where the magnitude of concept drift fluctuates.

Recognizing the importance of the window size, the data mining community has developed several adaptive algorithms that dynamically adjust the window size (Bifet and Gavalda, 2007; Deypir et al., 2012), and these techniques have also been used for clustering (Zhu et al., 2010). However, despite their success in practice, a formal theoretical understanding of the selection of the window size and its relation to the quality of the obtained solution remains largely unexplored. The aim of our work is to bridge this gap. Our main contribution lies in providing a rigorous mathematical analysis of the $(k, z)$-approximation problem with drifting data, establishing both an upper bound through a dynamic window size algorithm, and a matching lower bound for the problem. Our algorithm adapts to changing drift, computing a near-optimal window size at each step.

**Our Contribution**   We address the important question of how to approximate the current distribution of the data from a sequence of samples whose distribution is changing over time. Our work makes the following specific contributions:

1. We introduce a formal theoretical model to analyze the $(k, z)$-approximation problem for data streams with concept drift.

2. We introduce a novel technique to dynamically adjust the window size $w$ in response to the drift of the underlying distribution, and provide a theoretical analysis of the quality of the solution (Theorem 2). This is a challenging problem since, as we will see, it is not possible to quantify the drift from the input data. Despite this, we demonstrate that our method can adjust $w$ as effectively as any algorithm with a full knowledge of the drift.

3. Through a minimax lower bound, we establish that our approach strikes a near-optimal trade-off on the window size $w$ for the $(k, 2)$-approximation problem (Theorem 4). This lower bound also demonstrates that a properly implemented window size strategy can provide a near-optimal adaptation to drift, justifying previous empirical clustering approaches based on this strategy (Zhu et al., 2010).

Furthermore, we provide preliminary experimental evidence of the ability of our algorithm to adapt the window size according to the drift of the distribution.

**Organization of the paper**   The rest of the paper is structured as follows. Section 2 defines formally the problem setting and outlines the challenges in adapting to drift. Section 3 discusses our main theoretical contributions. Section 4 provides an account of the relevant previous work. Section 5 presents the algorithm and sketches its analysis, while Section 6 presents a sketch of the lower bound. Section 7 reports the results of the experiments. Section 8 concludes with some final remarks. Details of the proofs are deferred to the appendix.

## 2. Preliminaries

For a point $p \in \mathbb{R}^d$ and a set $S \subseteq \mathbb{R}^d$, we abuse notation and define the distance from $p$ to $S$ as $\|p - S\| \doteq \inf_{s \in S} \|p - s\|$, where $\|\cdot\|$ is the Euclidean norm. Given an integer $k \geq 1$, a real $z \geq 1$, and a set of $k$ centers $C \subset \mathbb{R}^d$, we remind that the definition of the cost function $\mathcal{W}_{k,z}(C, P)$ for the $(k, z)$-approximation problem is given by Equation (1). We also recall that the $(k, z)$-approximation of a distribution $P$ requires finding a set $C$ of at most $k$ points minimizing $\mathcal{W}_{k,z}(C, P)$. Also, the $(k, z)$-clustering problem of a finite multiset of points $S \subseteq \mathbb{R}^d$ is equivalent to the $(k, z)$-approximation problem of the uniform distribution over $S$.

We adopt the following data-generating model that has also been used in previous work on drift (Mohri and Muñoz Medina, 2012; Mazzetto and Upfal, 2023b). The input data stream is a sequence of random variables $(X_t)_{t \in Z^+}$ where $X_t$, with $t \geq 1$, is sampled from a distribution $P_t$ over $\mathbb{R}^d$, which represents the current state of the world at time step $t$. We assume that the random variables $\{X_t : t \in \mathbb{Z}^+\}$ are mutually independent, and that the support of each distribution is contained in the ball of radius $1/2$ centered in the origin (i.e., $\Pr(\|X_t\| \leq 1/2) = 1$ for all $t \geq 1$)[1].

Let $\mathcal{C}$ be the family of all possible sets of $k$ centers, that is,

$$\mathcal{C} = \left\{ \{c_1, \ldots, c_k\} \ : \ c_i \in \mathbb{R}^d, \|c_i\| \leq 1/2, \text{ for } 1 \leq i \leq k \right\} .$$

---

1. Note that by a simple rescaling, our results can be extended to any finite radius $r \in \mathbb{R}$.

We let $\mathcal{W}_{k,z}^*(P) \doteq \inf_{C \in \mathcal{C}} \mathcal{W}_{k,z}(C, P)$ be the minimum cost attainable for the $(k, z)$-approximation of $P$. Let $C^*$ be any optimal solution for $P$ with cost $\mathcal{W}_{k,z}^*(P)$. We remark that the value $\mathcal{W}_{k,z}^*(P)$ can also be interpreted as the quantization error of representing the distribution $P$ with the points $C^*$ (Graf and Luschgy, 2007; Gersho and Gray, 2012). For instance, if we want to represent the distribution $P$ using a single point ($k = 1$) with respect to the squared error ($z = 2$), then $C^*$ is the mean of $P$, and $\mathcal{W}_{k,z}^*(P)$ is the trace of the covariance matrix of $P$.

Let $T$ denote the current time. Our goal is to find a set of centers $C \in \mathcal{C}$ which minimizes $\mathcal{W}_{k,z}(C, P_T)$, thus providing an optimal solution to the $(k, z)$-approximation problem with respect to the current "state of the world" distribution $P_T$. One of the main challenges is that the distribution $P_T$ is unknown, thus we cannot directly compute the costs $\{\mathcal{W}_{k,z}(C, P_T) : C \in \mathcal{C}\}$, needed for the solution of the minimization problem. Since we have only *one single* sample $X_T$ from $P_T$, those costs must be estimated from past samples, provided that no or minimal drift occurred, i.e., the distribution of those samples closely resembles $P_T$. Nevertheless, accurately estimating drift is unfeasible, as one sample per distribution is available.

To further proceed with the discussion, we introduce additional notation. To quantify the drift, we define the following problem-specific distance $\tau$ between distributions. Given two distributions $P$ and $Q$ over $\mathbb{R}^d$, we let

$$\tau(P, Q) = \sup_{C \in \mathcal{C}} |\mathcal{W}_{k,z}(C, P) - \mathcal{W}_{k,z}(C, Q)| \ . \tag{2}$$

It is easy to verify that $\tau$ defines a pseudometric[2]. In the domain adaptation literature (Mohri and Muñoz Medina, 2012), the distance $\tau(P, Q)$ from $P$ to $Q$ is referred to as the *discrepancy* induced by the family of functions $\{x \mapsto \|x - C\|^2 : C \in \mathcal{C}\}$. The discrepancy provides a measure of the drift that is problem-dependent. It is possible to upper bound the discrepancy $\tau$ using more common metrics between distributions such as the Wasserstein distance (see Appendix A.2).

Let $\delta_x$ be the Dirac function centered at $x \in \mathbb{R}^d$. For a window size $1 \le w \le T$, we define

$$P_T^{[w]} = \frac{1}{w} \sum_{t=T-w+1}^{T} P_t, \qquad \mathbb{P}_T^{[w]} = \frac{1}{w} \sum_{t=T-w+1}^{T} \delta_{X_t} \ . \tag{3}$$

The distribution $P_T^{[w]}$ is the average of the distributions of the most recent $w$ samples. The distribution $\mathbb{P}_T^{[w]}$ is the empirical distribution induced by the latest $w$ samples, namely, the discrete distribution where each point $x \in \mathbb{R}^d$ has probability $c_x/w$, where $c_x$ is the number of realizations of $X_{T-w+1}, \ldots, X_T$ equal to $x$.

Our goal is to identify a window size $w$ for which the empirical distribution of the samples $\mathbb{P}_T^{[w]}$ is representative of $P_T$, that is, $\tau(P_T, \mathbb{P}_T^{[w]})$ is small, thus ensuring that the empirical costs computed from the last $w$ samples closely approximate the expected costs of solutions under the true distribution $P_T$. If we can find such a window size $w$, then we can obtain a solution for $P_T$ by using an algorithm for $(k, z)$-clustering over the most recent $w$ samples. To evaluate a window size $w$, we upper bound $\tau(P_T, \mathbb{P}_T^{[w]})$ based on an error decomposition into *statistical error* and *drift error*, which was used in previous work on drift (Mazzetto and Upfal, 2023a,b). The bound is stated in the following lemma (proof in Appendix B).

---

2. For a pseudometric $\tau$, it could be that $\tau(x, y) = 0$ even if $x \ne y$.

**Lemma 1** *For any $1 \le w \le T$, the following inequality holds*

$$\tau(P_T, \mathbb{P}_T^{[w]}) \le \underbrace{\tau(P_T^{[w]}, \mathbb{P}_T^{[w]})}_{\text{statistical error}} + \underbrace{\max_{0 \le t < w} \tau(P_T, P_{T-t})}_{\text{drift error}} \quad .$$

The statistical error quantifies the error caused by the stochasticity of the data-generating process, while the drift error is due to the non-stationarity of the data-generating process. Intuitively, there is a trade-off between those two errors: larger window sizes $w$ yield smaller statistical errors but may incur larger drift errors since we use samples from distributions that are further away from the current distribution.

There are different challenges in the solution of this trade-off. As previously noted, we cannot estimate the drift error since we only have access to a single sample from each distribution. Thus, it is not possible to analytically solve the trade-off to determine the optimal window size. Additionally, it is computationally hard to calculate the metric $\tau$ from the data. The computation of the distance $\tau$ between two empirical distributions requires evaluating the supremum over all possible sets of $k$ centers in $\mathcal{C}$. This is a computationally hard problem: for $k = 1$ and $z = 1$, this is equivalent to the Fermat-Weber problem with attraction and repulsion, for which only approximate solutions are available (e.g., Brimberg (1995)), and no solutions are known for $k > 1$.

A further obstacle is that even if we can determine the window size $w$ to use, there is no efficient algorithm for computing an optimal solution to the $(k, z)$-clustering problem (this problem is known to be NP-hard at least for $z \in \{1, 2\}$, see the work of Dasgupta (2008) and Cohen-Addad et al. (2022)). Instead, we assume that a $\beta$-approximation algorithm to the $(k, z)$-clustering problem is available. Such an algorithm computes a solution whose cost is bounded by $\beta$ times the cost of the optimal solution to the problem, where $\beta \ge 1$ is also referred to as *approximation factor*. We aim to ensure that, despite the algorithm facing multiple complex decisions to tackle drift, we can still achieve the same multiplicative error of the best-known approximation for the considered $(k, z)$-clustering problem in the no-drift scenario.

## 3. Main Result

Our work has two primary contributions. Firstly, we show that there exists an algorithm that provides a solution whose cost guarantee is nearly as good as that of a solution computed using a window size that minimizes the trade-off between statistical error and drift error (Theorem 2). Secondly, we exhibit a lower bound for the special case of $k$-means ($z = 2$) that demonstrates that our algorithm provides a guarantee that is near-optimal in a drift setting (Theorem 4).

### 3.1. Upper Bound.

The next theorem characterizes our algorithm's effectiveness in adapting to drift.

**Theorem 2** *Let $\delta \in (0, 1)$, and assume that we have access to a $\beta$-approximation algorithm for the $(k, z)$-clustering problem. There exists an algorithm that observes $X_1, \dots, X_T$ and outputs a solution $\hat{C}$ such that, with probability at least $1 - \delta$, it holds*

$$\mathcal{W}_{k,z}(\hat{C}, P_T) \le \beta \mathcal{W}_{k,z}^*(P_T) + \beta \cdot O\left(\min_{1 \le w \le T}\left[\sqrt{\frac{z^2 k \log^4(w) + \log(1/\delta)}{w}} + \max_{0 \le t < w} \tau(P_T, P_{T-t})\right]\right).$$

To appreciate this bound, suppose that we fix a window size $w \in \{1, \dots, T\}$ arbitrarily, and that we compute a $\beta$-approximation solution $\tilde{C}_T^{[w]}$ for the $(k, z)$-clustering problem over the latest $w$ samples $X_{T-w+1}, \dots, X_T$, i.e., $\mathcal{W}_{k,z}(\tilde{C}_T^{[w]}, \mathbb{P}_T^{[w]}) \leq \beta \mathcal{W}_{k,z}^*(\mathbb{P}_T^{[w]})$. The following lemma (proof in Appendix B) provides an upper bound on the cost of such a solution with respect to the current distribution $P_T$.

**Lemma 3** *For any $1 \leq w \leq T$, the following inequality holds*

$$\mathcal{W}_{k,z}(\tilde{C}_T^{[w]}, P_T) \leq \beta \cdot \mathcal{W}_{k,z}^*(P_T) + (\beta + 1)\Big[ \underbrace{\tau(P_T^{[w]}, \mathbb{P}_T^{[w]})}_{\text{statistical error}} + \underbrace{\max_{0 \leq t < w} \tau(P_T, P_{T-t})}_{\text{drift error}} \Big] \ .$$

The upper bound of Lemma 3 is interpreted as follows. The first term $\beta \cdot \mathcal{W}_{k,z}^*(P_T)$ represents the multiplicative error introduced by using a $\beta$-approximation algorithm, and it is independent to $w$ (we remind that $\mathcal{W}_{k,z}^*(P_T)$ is the cost of the optimal solution for $P_T$). The other term depicts the additive error introduced by the trade-off between the statistical error and drift error (see Lemma 1).

The tightest upper bound of Lemma 3 is achieved by choosing a window size $w$ that optimally solves the trade-off between statistical error and drift error. Theorem 2 shows that we can successfully approach the optimal solution of this trade-off (min in the upper bound), where the statistical error is upper bounded by $\tilde{O}(\sqrt{z^2 k/w})$. This upper bound to the statistical error can be proven to be tight up to logarithmic factors (Bartlett et al., 1998) for the $k$-means problem ($z = 2$). The additional $O(\sqrt{\log(1/\delta)/w})$ term is due to the high-probability guarantee. We remark that our algorithm does not require any prior knowledge of the drift, and it overcomes the challenge that the drift error *cannot* be estimated from the input data.

Additionally, the proof of Theorem 2 also requires a novel upper bound on the statistical error of the $(k, z)$-approximation problem for arbitrary $z \geq 1$, extending existing results that only hold for constant $z$ (Bucarelli et al., 2023).

Significantly, our result also features a multiplicative error term proportional to exactly $\beta$, with $\beta$ being the best approximation algorithm for the $(k, z)$-clustering problem. Although our algorithm involves multiple steps and invocations of a $\beta$-approximation algorithm, the final approximation factor remains unaffected.

We emphasize that our algorithm dynamically chooses the window size in order to minimize the trade-off between statistical error and drift error. If the distribution changes arbitrarily and drifts significantly every few steps, learning becomes infeasible, and even the optimal trade-off would yield a poor solution. Also, note that our algorithm makes no prior assumptions about the structure of the drift and can handle arbitrary changes in the distribution. An important special case is gradual drift, where the change in distribution between consecutive steps is bounded; this case is discussed in more detail in Section 3.2. Another important case is abrupt drift, where the input sequence can be divided into long distinct segments, with the distribution remaining constant within each segment but potentially varying significantly across different segments. In such cases, the optimal solution of the trade-off between statistical error and drift error, as outlined in Theorem 2, would lead to selecting samples only from the latest segment. A visual representation of the window sizes selected by the algorithm across these two different drift regimes is provided in Section 7.

### 3.2. Lower Bound.

We show the tightness of our upper bound for the classical $(k, 2)$-approximation problem ($z = 2$). We consider a classical setting where there is a bounded drift at each step, which is widely used in the literature on learning with drift (Bartlett, 1992; Mohri and Muñoz Medina, 2012). Formally, we assume that there exists $\Delta > 0$ such that $\tau(P_{t+1}, P_t) \leq \Delta$ for any $t \geq 1$, which immediately implies that $\max_{t<w} \tau(P_T, P_{T-t}) \leq w\Delta$. Thus, by choosing $w = (k/\Delta^2)^{1/3}$ in Theorem 2, we have that with high-probability (e.g., $\geq 0.99$), our algorithm returns a solution $\hat{C}$ such that

$$\mathcal{W}_{k,z}(\hat{C}, P_T) \leq \beta \mathcal{W}_{k,z}^*(P_T) + \beta \cdot \tilde{O}\left((k\Delta)^{1/3}\right) \quad . \tag{4}$$

We remark that our algorithm achieves this guarantee without knowing the value $\Delta$ a priori.

In our lower bound, we show that the upper bound (4) is tight up to logarithmic factors, and it cannot be improved without further assumptions. The following result follows by combining lower bound techniques for vector quantization (Bartlett et al., 1998) and learning with drift (Mazzetto and Upfal, 2023b).

**Theorem 4** *Let $k \geq 3$, $z = 2$, and assume $\Delta \in (0, 1/k^{1+1/d})$ and $T \geq (k^{1-2/d}/\Delta^2)^{1/3}$. Consider an algorithm that observes $X_1, \ldots, X_T$, and outputs $k$ centers $\hat{C} \in \mathcal{C}$. For any such algorithm, there exists a sequence of distributions $P_1, \ldots, P_T$ such that $\tau(P_t, P_{t+1}) \leq \Delta$ for any $1 \leq t \leq T - 1$ and*

$$\mathbb{E}\,\mathcal{W}_{k,z}(\hat{C}, P_T) \geq \mathcal{W}_{k,z}^*(P_T) + \Omega\left(k^{\frac{1}{3}[1-\frac{5}{d}]}\Delta^{1/3}\right) \quad .$$

A sketch of the proof is provided in Section 6, while the full proof is deferred to Appendix C. For $\beta \to 1$ and $d \to \infty$, we observe that the upper bound (4) matches up to logarithmic factors the lower bound in expectation of Theorem 4. This result highlights the tightness of our approach in a drift setting.

We wish to remark that, aside from polylogarithmic factors, there remains a polynomial gap in the dependence on $k$ between the lower and the upper bound, for constant $d$. This gap is not unique to our analysis, and it also appears in the best-known lower and upper bounds for the setting with independent and identically distributed samples (Bartlett et al., 1998). Indeed, bridging this gap remains an interesting open question.

## 4. Related Work

The $(k, z)$-clustering problem, along with its popular variants $k$-median and $k$-means, has been extensively studied in the data stream model (see surveys by Silva et al. (2013) and Aggarwal (2018)). A first line of research focused on obtaining good solutions with respect to all the past points in a streaming setting (Ailon et al., 2009; Charikar et al., 2003; Chen, 2006; Braverman et al., 2011; Barger and Feldman, 2016; Cohen-Addad et al., 2023). Different methodologies have been proposed to tackle the concept drift. A line of work developed efficient algorithms that maintain a solution over a sliding window of fixed size (Babcock et al., 2003; Braverman et al., 2015, 2016; Borassi et al., 2020; Epasto et al., 2022; Woodruff et al., 2024). Another alternative is to use a decaying weighting of the past samples (Bidaurrazaga et al., 2021). These works focus on the computational efficiency of maintaining such solutions given an arbitrarily fixed window size or decay parameter. In contrast, we theoretically characterize the quality of the solution for a given window size and propose an *adaptive* method that adjusts the window size based on a simple statistical interpretation

of the data points. While beyond the scope of this paper, it is worth noting that the literature has explored several clustering variants beyond center-based methods (we refer interested readers to the book by Hennig et al. (2015)).

The idea of using sliding windows of adjustable length to adapt to time-varying drift has been empirically explored for clustering (Zhu et al., 2010), and for other data mining problems (e.g., Bifet and Gavalda, 2007; Deypir et al., 2012; Jiang and Zhang, 2004; Kumar and Satapathy, 2014). Unlike these works, we provide a rigorous analysis of the quality of the solution obtained by our algorithm.

There is a vast literature on the problem of concept drift in learning and data mining (Agrahari and Singh, 2022; Gama et al., 2014; Lu et al., 2018). We review the work that is closer to our setting, where the input is modeled as a sequence of independent samples each originating from a different distribution. This drift setting has been widely studied in the learning theory literature, and a first line of work addressed the problem of learning a family of binary classifiers (Helmbold and Long, 1991; Bartlett, 1992; Helmbold and Long, 1994; Barve and Long, 1996; Long, 1998). This analysis was later extended to handle arbitrary family of functions (Mohri and Muñoz Medina, 2012), and to relax the independence assumption using a mixing assumption (Hanneke and Yang, 2019). These works all assume that an upper bound to the magnitude of the drift is known a priori, which is used to quantify the drift error. In the binary and realizable setting, an algorithm that is adaptive to the drift was proposed by Hanneke et al. (2015). More recent work extends this adaptive result to a general learning setting (Mazzetto and Upfal, 2023a). The latter work does not apply to our setting as it requires the computation of the discrepancy (the distance $\tau$ defined in (2)), which is computationally unfeasible for the $(k, z)$-approximation problem. The drift setting has also been considered for other important problems such as density estimation (Mazzetto and Upfal, 2023b; Mazzetto, 2024), weak supervision (Fu et al., 2020; Mazzetto et al., 2025), or model selection (Han et al., 2024).

Another rich area of research for adapting to adversarially changing environments is online learning, where the goal is to minimize regret over a long sequence of predictions with respect to the best solution in hindsight (e.g., see Shalev-Shwartz et al. (2012); Hazan et al. (2016); Orabona (2019)). Recently, the regret of the $k$-means problem has been studied in this setting (Cohen-Addad et al., 2021a), where at each time step $t$, the adversary can choose the point $X_t$ after the algorithm determines a solution $C_t$.

Finally, it is important to note that the statistical error of the $(k, z)$-approximation problem has been widely studied in the literature for the setting of independent and identically distributed samples. A first line of work established upper bounds for the statistical error of the $(k, 2)$-approximation problem, which is also referred to as vector quantization, or simply $k$-means (Bartlett et al., 1998; Linder, 2000; Antos et al., 2005; Biau et al., 2008; Fefferman et al., 2016; Liu, 2021; Appert and Catoni, 2021). An almost matching minimax lower bound for the statistical error is also known for the $(k, 2)$-approximation problem (Bartlett et al., 1998). Subsequent research provided an upper bound to the statistical error of the $(k, z)$-approximation problem for a constant $z$ (Bucarelli et al., 2023). Additionally, the statistical error in other clustering variants such as kernel $k$-means (Liu, 2021), spectral clustering (Li and Liu, 2021), and projective clustering (Bucarelli et al., 2023) has also been explored.

## 5. Sketch of the Upper Bound

In this section, we present the algorithm that attains the guarantee of Theorem 2. The formal analysis and a more detailed discussion are deferred to Appendix B. As previously noted, the major difficulty

is that it is not possible to obtain an accurate estimate of the drift error $\max_{0 \le t < w} \tau(P_T, P_{T-t})$ from the data, as we have access to only one sample from each distribution, and it is computationally intractable to evaluate the metric $\tau$. Thus, our algorithm needs to choose a window size that yields an optimal solution of the trade-off between statistical error and drift error without explicitly estimating the drift error. Another challenge is that the algorithm can only rely on approximate clustering solutions whose cost can deviate from the optimal arbitrarily within their approximation guarantee.

Our algorithm employs a *lazy strategy*, iteratively increasing the window size until evidence of drift is detected. The key intuition is as follows. Let $T$ be the current time, and let $\tilde{C}$ be an approximate solution for the $(k, z)$-clustering problem over the latest $w$ samples. We compare $\tilde{C}$ with another solution $\tilde{C}'$ obtained using $w' > w$ samples. If there is no drift, the costs $\mathcal{W}_{k,z}(\tilde{C}, \mathbb{P}_T^{[w]})$ and $\mathcal{W}_{k,z}(\tilde{C}', \mathbb{P}_T^{[w]})$ should not differ beyond the approximation error and the statistical error due to finite sample size $w$. If these costs are similar, we prefer the solution $C'$ since it exhibits a smaller statistical error, and we continue to increase the window size. Conversely, if the costs are significantly different, it indicates that the data are not all sampled from the same distribution, providing evidence that a drift occurred. In this case, we can establish a lower bound on the drift error using $w'$ samples, which is sufficient to stop iterating. On a high-level, Theorem 2 is derived by carefully determining the threshold for comparing the solutions $\tilde{C}$ and $\tilde{C}'$.

Formally, the algorithm operates as follows. Let $T$ be the current time. Our algorithm considers windows of doubling sizes $1, 2, 4, \ldots,$, and we let $w_j = 2^j$ be the window size considered at iteration $j$, starting from $j = 0$. There is nothing special about using powers of 2, and this choice is made only for ease of presentation, since our algorithm can be easily extended to consider any power of $\gamma > 1$ Let $\delta \in (0, 1)$ be the failure probability of our algorithm. The algorithm uses the following upper bound $G(w_j)$ to the statistical error with $w_j$ samples: with probability at least $1 - \delta$, it holds that

$$\forall j \in \mathbb{Z}^+, \quad G(w_j) \doteq c \left[ \sqrt{\frac{z^2 k \log^4(w) + \log(1/\delta)}{w_j}} \right] \ge \tau(P_T^{[w_j]}, \mathbb{P}_T^{[w_j]}) \ , \tag{5}$$

where $c \ge 0$ is a universal constant. The above upper bound is obtained by extending existing results on the statistical error known for constant $z$ to an arbitrary finite $z \ge 1$ (Biau et al., 2008; Maurer, 2016; Liu, 2021; Bucarelli et al., 2023) (see Appendix B.1 for details). We remark that our approach does not require the use of this specific upper bound, and it is trivial to adapt the results to any upper bound on the statistical error, as long as the bound is polynomially decreasing with $w$.

The algorithm has access to a $\beta$-approximation algorithm $\mathcal{A}$ for the $(k, z)$-clustering problem. At the beginning of iteration $j$, we compute a solution $\mathcal{A}(\mathbb{P}_T^{[w_j]}) \in \mathcal{C}$ for the $(k, z)$-clustering problem over the points $X_{T-w_j+1}, \ldots, X_T$, such that $\mathcal{W}_{k,z}(\mathcal{A}(\mathbb{P}_T^{[w_j]}), \mathbb{P}_T^{[w_j]}) \le \beta \cdot \mathcal{W}_{k,z}^*(\mathbb{P}_T^{[w_j]})$. For technical reasons, we define $\tilde{C}_T^{[w_j]}$ as the solution that minimizes the cost computed with respect to the empirical distribution $\mathbb{P}_T^{[w_j]}$ among all approximate solutions computed by $\mathcal{A}$ on windows of size $w_i$ with $0 \le i \le j$. That is, $\tilde{C}_T^{[w_j]} = \operatorname{argmin}\left\{\mathcal{W}_{k,z}\left(C, \mathbb{P}_T^{[w_j]}\right) : C \in \{\mathcal{A}(\mathbb{P}_T^{[w_i]})\}_{0 \le i \le j}\right\}$. Note that since $\beta > 1$, it is possible that a set of centers computed with respect to another smaller window size $w_i < w_j$ provides a better solution for $\mathbb{P}_T^{[w_j]}$. This definition of $\tilde{C}_T^{[w_j]}$ is used to reduce the effect of the error due to the approximation of the algorithm when comparing the solutions across different window sizes.

---

**Algorithm 1:** Adaptive $(k, z)$-approximation at time step $T$

---

$\tilde{C}_T^{[w_0]} \leftarrow \mathcal{A}(\mathbb{P}_T^{w_0})$

$\mathcal{S} \leftarrow \{\mathcal{A}(\mathbb{P}_T^{w_0})\}$

**for** $j = 1, 2, \ldots, \lfloor \log_2(T) \rfloor$ **do**

    $\mathcal{S} \leftarrow \mathcal{S} \cup \{\mathcal{A}(\mathbb{P}_T^{w_j})\}$

    $\tilde{C}_T^{[w_j]} \leftarrow \mathrm{argmin}_{C \in \mathcal{S}} \, \mathcal{W}_{k,z}\left(C, \mathbb{P}_T^{[w_j]}\right)$

    **for** $i = 0, \ldots, j-1$ **do**

        **if** $\mathcal{W}_{k,z}(\tilde{C}_T^{[w_j]}, \mathbb{P}_T^{[w_i]}) - \mathcal{W}_{k,z}(\tilde{C}_T^{[w_i]}, \mathbb{P}_T^{[w_i]}) \geq 6G(w_j) + 2G(w_i)$ **then**

            **return** $\tilde{C}_T^{[w_{j-1}]}$

        **end**

    **end**

**end**

**return** $\tilde{C}_T^{[w_{\lfloor \log_2(T) \rfloor}]}$

---

The pseudocode of the algorithm is reported in Algorithm 1. The proof of Theorem 2, provided in full in Appendix B, follows the key intuition presented at the beginning of this section. First, we will show that if at iteration $j$ the `if` condition is false for all $i = 0, \ldots, j-1$, then the guarantee provided by the solution $\tilde{C}_T^{[w_j]}$ is up to constants as good as the one provided by any of the previous solutions. In this case, our algorithm keeps iterating. Second, we will show that if at iteration $j$ there exists $i < j$ such that the `if` condition is true, then a sizeable distribution drift occurred from $P_{T-w_j+1}$ to $P_T$. This is the crucial technical result that allows the algorithm to lower bound the drift without explicitly computing it. In particular, we will show that when this occurs, then $\max_{0 \leq t < w_j} \tau(P_T, P_{T-t}) \geq G(w_j)$. Since the drift error is monotonically increasing with $w$, the drift error will dominate over the statistical error for any $w \geq w_j$: this is sufficient to interrupt the iteration loop of our algorithm. Then, the proof of the theorem will be constructed using those two results.

**Running time and memory.** At every time $T$, under the assumption that all points of the stream $X_1, X_2, \ldots, X_T$ are available in memory, Algorithm 1 must compute at most the $O(\log T)$ solutions and $O(\log^2 T)$ costs. For concreteness, let $z = 2$. If the solutions are computed using $k$-means++ followed by a constant number of iterations of Lloyd's algorithm (Arthur and Vassilvitskii, 2007), the overall running time would be $O(kT \log T)$. It is important to remark, that when $T$ grows very large, the performance of the algorithm can be drastically improved by limiting the window sizes to a maximum length $L$, which would then replace $T$ in the time-bound. In this case, it is easy to argue that the upper bound of Theorem 2 stays the same, aside from an extra error term $O(\beta G(L))$, where $G(L)$ bounds the statistical error with $L$ samples. Thus, if we want to ensure that this extra error is $O(\epsilon)$, we can limit the algorithm to consider window sizes up to length $L = G^{-1}(\epsilon/\beta)$. In this case, it is also sufficient to only keep in memory the most recent $O(L)$ samples at each time step. Further performance improvements can be obtained by leveraging the techniques presented by Epasto et al. (2022) and Woodruff et al. (2024) for $k$-means clustering in the sliding window setting, by maintaining only a sketch of every window, incurring, however, a slight worsening of the approximation.

## 6. Sketch of the Lower Bound

In this section, we provide the main intuition behind the proof of the lower bound of Theorem 4. The full details of the proof are deferred to Appendix C. Let $T$ and $\Delta$ satisfy the assumption of the theorem. We remind that the goal is to show that for any algorithm that takes in input a sequence $X_1, \ldots, X_T$ from $\mathbb{R}^d$ and outputs a set of at most $k$ centers $\hat{C}$, there exists a sequence of distributions $P_1, \ldots, P_T$ of the random variables $X_1, \ldots, X_T$ such that $\tau(P_t, P_{t+1}) \leq \Delta$ for each $t \in \{1, \ldots, T-1\}$, and

$$\mathbb{E} \, \mathcal{W}_{k,2}(\hat{C}, P_T) - \mathcal{W}_{k,2}^*(P_T) = \Omega\left(k^{\frac{1}{3}[1-\frac{5}{d}]}\Delta^{1/3}\right) \ .$$

We provide a sketch of the proof for the simpler case where $d = 1$ and $k = 3$. To establish the lower bound, we reformulate the problem as a decision problem based on two possible distributions of the input sequence $X_1, \ldots, X_T$.

In particular, we consider distributions with support over four points $\{x_1, x_2, y_1, y_2\} \subseteq \mathbb{R}$, where $x_1 = 1/2$, $x_2 = 1/2 - \gamma$, $y_1 = -1/4$ and $y_2 = -1/4 + \gamma$. Let $\xi \in \{-1, 1\}$. We consider two distributions over $\boldsymbol{X} = (X_1, \ldots, X_T)$ parameterized by $\xi$: the product distribution $\boldsymbol{P}^{[1]} = \bigotimes_{t=1}^T P_t^{[1]}$ and the product distribution $\boldsymbol{P}^{[-1]} = \bigotimes_{t=1}^T P_t^{[-1]}$. Fix $w \in \{1, \ldots, T\}$, we let

$$P_t^{[\xi]}(z) = \begin{cases} \frac{1}{4} + \frac{\xi\Delta}{2}[t - T + w]_+ & \text{if } z \in \{x_1, x_2\} \\ \frac{1}{4} - \frac{\xi\Delta}{2}[t - T + w]_+ & \text{if } z \in \{y_1, y_2\} \end{cases} ,$$

where $[x]_+ \doteq \max\{0, x\}$. For both sequences, the distribution of the first $T - W$ random variables is uniform over $\{x_1, x_2, y_1, y_2\}$. For the last $w$ random variables, if $\xi = 1$, their distribution will have increasingly larger probability mass to $\{x_1, x_2\}$, and vice-versa for $\xi = -1$. It is easy to see that for any $t$, the drift is upper bounded $\tau(P_t, P_{t+1}) \leq \Delta$ (note that all distances are $\leq 1$).

By construction, the optimal centers for $P_T^{[1]}$ are $C^{[1]} \doteq \{x_1, x_2, (y_1 + y_2)/2\}$, whereas the optimal centers for $P_T^{[-1]}$ are $C^{[-1]} = \{y_1, y_2, (x_1 + x_2)/2\}$. Additionally, if $\gamma \leq 1/16$ and $w\Delta \leq 1/16$, one can easily verify that for any set of $k$ centers $C$, there is a $\hat{C} \in \{C^{[-1]}, C^{[1]}\}$ such that $\mathcal{W}_{3,2}(\hat{C}, P_T^{[\xi]}) \leq \mathcal{W}_{3,2}(C, P_T^{[\xi]})$ for $\xi \in \{-1, 1\}$ (To check this, it is sufficient to consider solutions $C$ constructed as follows: partition $\{x_1, x_2, y_1, y_2\}$ into 3 sets, and find the optimal center for each set). Thus, to establish the desired result, we need only to provide a lower bound to the following quantity:

$$\begin{aligned} \Psi &\doteq \inf_{\hat{C}} \sup_{\xi \in \{-1,1\}} \mathbb{E}_{\boldsymbol{X} \sim \boldsymbol{P}^{[\xi]}} \left(\mathcal{W}_{3,2}(\hat{C}, P_T^{[\xi]}) - \mathcal{W}_{3,2}^*(P_T^{[\xi]})\right) \\ &= \inf_{\hat{C} \in \{C^{[-1]}, C^{[1]}\}} \max_{\xi \in \{-1,1\}} \mathbb{E}_{\boldsymbol{X} \sim \boldsymbol{P}^{[\xi]}} \left(\mathcal{W}_{3,2}(\hat{C}, P_T^{[\xi]}) - \mathcal{W}_{3,2}^*(P_T^{[\xi]})\right) \ . \end{aligned} \tag{6}$$

Equation (6) shows that we can obtain a lower bound by formulating a decision problem of determining $\xi$ given the observations $X_1, \ldots, X_T$. Let $\hat{\phi}$ be any estimator that observes $\boldsymbol{X}$ and it decides whether $\xi = 1$ or $\xi = -1$. The optimal $(3, 2)$-approximation solution $C^*$ for $P_T^{[\xi]}$ has cost $\frac{\gamma^2}{2}\left(\frac{1}{2} - w\Delta\right)$, but if this solution is used to approximate $P_T^{[-\xi]}$, the cost is equal to $\frac{\gamma^2}{2}\left(\frac{1}{2} + w\Delta\right)$. Therefore, it holds:

$$\Psi = \gamma^2 w\Delta \cdot \inf_{\hat{\phi}} \max_{\xi \in \{-1,1\}} \mathbb{E}_{\boldsymbol{X} \sim \boldsymbol{P}^{[\xi]}} \mathbf{1}_{\{\hat{\phi}(\boldsymbol{X}) \neq \xi\}} \ . \tag{7}$$

where $\mathbf{1}_A \in \{0, 1\}$ is an indicator function that is equal to 1 if and only if $A$ is true. We can use an information-theoretic lower bound as Le Cam's lower bound (Lemma 1 of Yu, 1997) to obtain

$$\inf_{\hat{\phi}} \max_{\xi \in \{-1,1\}} \mathbb{E}_{\boldsymbol{X} \sim \boldsymbol{P}^{[\xi]}} \mathbf{1}_{\{\hat{\phi}(\boldsymbol{X}) \neq \xi\}} \geq \frac{1}{2} \left( 1 - \frac{1}{2} \|\boldsymbol{P}^{[1]} - \boldsymbol{P}^{[-1]}\|_1 \right) \geq \frac{e^{-\mathrm{KL}\left(\boldsymbol{P}^{[1]}, \boldsymbol{P}^{[-1]}\right)}}{4} \qquad (8)$$

where KL denotes the Kullback–Leibler divergence, and the last inequality is folklore (e.g., Tsybakov, 2008). By direct computation, the KL divergence between the two distributions is upper bounded as

$$\mathrm{KL}\left( \boldsymbol{P}^{[1]}, \boldsymbol{P}^{[-1]} \right) = \sum_{t=1}^{T} \mathrm{KL}\left( \boldsymbol{P}_t^{[1]}, \boldsymbol{P}_t^{[-1]} \right) = O\left( w^3 \Delta^2 \right) \quad . \qquad (9)$$

By combining Equations (7), (8) and (9), we finally obtain that:

$$\Psi \geq \gamma^2 w \Delta e^{-O\left(w^3 \Delta^2\right)} \quad . \qquad (10)$$

Note that the value of $w$ represents a trade-off: for a larger $w$, we incur a larger error proportional to $\gamma^2 w$ if we decide incorrectly, however, it is easier to take the correct decision since the two distributions drifted apart for $w$ steps. By taking $\gamma = 1/16$ and $w = (1/\Delta)^{2/3}$ to maximize the lower bound of Equation (10), we finally obtain that $\Psi = \Omega(\Delta^{1/3})$.

To generalize this proof to an arbitrary $k \geq 3$, we construct a decision problem that contains $k/3$ rescaled instances of the decision problem for $k = 3$ formulated above. These instances must be sufficiently separated in space so that they can be treated independently. The dependency on the dimension $d$ arises from a covering argument, which determines the scaling required to ensure that all the instances fit within a ball of radius $1/2$.

A similar construction to formulate a decision problem to lower bound the error of the $(k, 2)$-approximation problem with independent and identically distributed samples was used by Bartlett et al. (1998). The extension of the lower bound to take into account a drift in the distribution is inspired by previous work on learning with drift (Mazzetto and Upfal, 2023b).

## 7. Simulations

We study the behavior of Algorithm 1 with a proof-of-concept implementation, limited to the case $z = 2$ (related to the $k$-means objective). We consider synthetic datasets generated using a mixture of four multivariate Gaussians in two dimensions, where drift is implemented by changing the mean and covariance matrix $\Sigma = I_2 \sigma^2$. Initially, each coordinate of the means is sampled uniformly from the interval $[-10, 10]$ (data will be rescaled later). We consider two datasets. SEGMENTED is a dataset where the stream is divided in four segments of length 256, 512, 128, and 256. Each segment maintains $\sigma$ constant ($\sigma = 1, 2, 1$, and $0.5$), and the means are moved at distance 10 in a random direction at the beginning of each segment. In this setting, the optimal window length at any time step $t$ extends from $t$ back to the beginning of the segment containing $t$. In the STEPWISE dataset, the stream is divided in three periods of 4096 points each, with drift occurring at each time step. In the first and last periods, $\sigma = 1$ and at each time step the means of the Gaussians are shifted by 0.05 along the first coordinate. In the second period, we set $\sigma = 2$, but the means are shifted by 0.001 at each time step. After generating the stream, we rescale all the points so that they are contained in the unit-norm ball. For both datasets, we report the length of the window selected by our algorithm at each time step, chosen among window sizes with geometric ratio $\gamma = 1.1$.

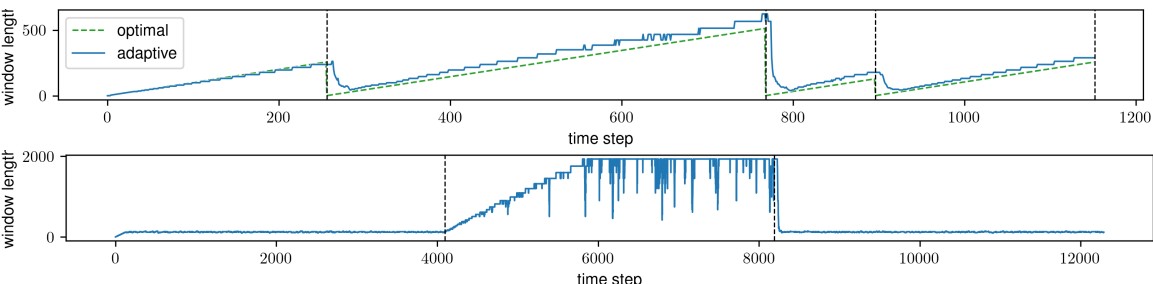

Figure 1: Window length at each time step on synthetic data: SEGMENTS (top) and STEPWISE (bottom).

For the SEGMENTED dataset (Figure 1, top), we observe that our algorithm (solid line) closely tracks the optimal window length (dashed line) across all segments. Vertical bars mark drift points. For the STEPWISE dataset (Figure 1, bottom) the vertical lines mark the boundary between the different periods. In this regime, we expect the window size to grow until we reach a good solution to the trade-off between statistical error and drift error. This behaviour is confirmed in all three segments. As expected, in the first and last segments, where considerable drift occurs at each time step, our algorithm uses short window lengths. The middle segment has a smaller drift but higher variance, resulting in a statistical error higher than the drift error. Therefore, our algorithm selects longer window lengths.

## 8. Conclusions

We address the $(k, z)$-approximation problem for a stream of data whose distribution can change over time. We present a variable-size window algorithm that adjusts its window length to provably adapt to the concept drift, and we show the tightness of our approach through a minimax lower bound for the important special case of $z = 2$. We believe that our framework is of more general interest, beyond the $(k, z)$-approximation problem, and can be applied to other important data mining and machine learning problems. As an example, it is possible to extend our work to other clustering variants such as spectral clustering or kernel $k$-means (Schölkopf et al., 1998; Ng et al., 2001; Li and Liu, 2021).

More generally, the framework works for any real-valued family of functions $\mathcal{F}$, where the cost $\mathcal{W}(f, P) = \int f(x)dP(x)$ is defined as the expectation of the function $f \in \mathcal{F}$, and $\tau(P, Q)$ is the discrepancy between $P$ and $Q$ according to the family $\mathcal{F}$. (For the $(k, z)$-approximation problem in this paper, we used $\mathcal{F} = \{x \mapsto \|x - C\|^z : C \in \mathcal{C}\}$.) There are only two requirements. The first is that one can compute the function with minimum cost, i.e., the empirical risk minimizer over a set of samples, but an approximate solution to this minimization problem also suffices. The second requirement is that we can provide an upper bound to the statistical error, i.e., a sample complexity result for the uniform convergence of the expected costs of $\mathcal{F}$. Our result improves on previous adaptive work on learning with drift that requires an exact computation of the function with the minimum cost, and also the additional assumption that the computation of the discrepancy $\tau$ is tractable (Mazzetto and Upfal, 2023a).

**Acknowledgments.** We thank the anonymous reviewers for their valuable feedback and suggestions, which helped improve the quality of this work. This work was partially supported by MUR of Italy, under Projects PRIN 2022TS4Y3N - EXPAND, and PNRR CN00000013 (National Centre for HPC, Big Data and Quantum Computing), and by a Kanellakis Fellowship.

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

## Appendix A. Auxiliary Results

In this section, we provide technical results that are useful to prove the main results of this paper.

### A.1. Useful Inequalities

**Proposition 5** *Let $a$ and $b$ be two real numbers, and let $z \geq 1$. We have:*

$$(a+b)^z \leq 2^{z-1}(a^z + b^z) \ .$$

**Proof** *This result is folklore. It holds:*

$$(a+b)^z = 2^z \left(\frac{a}{2} + \frac{b}{2}\right)^z \leq 2^z \left(\frac{a^z}{2} + \frac{b^z}{2}\right) = 2^{z-1}(a^z + b^z) \ ,$$

*where the inequality is due to the convexity of $q \mapsto q^z$ for $z \geq 1$.* ■

**Proposition 6 (Neta (1980))** *Let $z \geq 1$, and let $a$ and $b$ be two real numbers. We have:*

$$|a^z - b^z| \leq \max\left(\frac{z}{2}, 1\right) \cdot (a^{z-1} + b^{z-1}) \cdot |a - b|$$

### A.2. Relation between discrepancy and Wasserstein distance

Let $P$ and $Q$ be two distributions over $\mathbb{R}^d$. A joint distribution $\phi$ over $(\mathbb{R}^d)^2$ is a coupling for two distributions $P$ and $Q$ over $\mathbb{R}^d$ if $\int \phi(x,y)dx = Q(y)$ and $\int \phi(x,y)dy = P(x)$. The Wasserstein distance $\boldsymbol{W}(P,Q)$ between $P$ and $Q$ is defined as

$$\boldsymbol{W}(P,Q) = \inf_\phi \iint \|x - y\|\phi(x,y)dxdy \ .$$

We say that a coupling $\phi^*$ minimizing the above expression is an optimal coupling for $P$ and $Q$. Intuitively, $\phi^*$ is an optimal way to transport the probability mass from $P$ to $Q$ with respect to the Euclidean norm. The Wasserstein distance is often used to describe the distance between distributions for the $(k, 2)$-approximation problem (e.g., Zhuang et al., 2022).

**Lemma 7 (Relation between discrepancy and Wasserstein distance)** *Let $P$ and $Q$ be two distributions over $\mathbb{R}^d$ with support within the ball of radius $1/2$ centered at the origin. Let $\tau$ be the discrepancy as defined in Equation 2 for the $(k, z)$-approximation problem. The following inequality holds:*

$$\tau(P,Q) \leq \max(z, 2) \cdot \boldsymbol{W}(P,Q) \ .$$

**Proof** Let $\phi^*$ be the optimal coupling for $P$ and $Q$. Consider any $C \in \mathcal{C}$. We have that

$$\begin{aligned}
&\mathcal{W}_{k,z}(C, P) - \mathcal{W}_{k,z}(C, Q) \\
&= \int \|x - C\|^z P(x)dx - \int \|y - C\|^z Q(y)dy = \iint (\|x - C\|^z - \|y - C\|^z)\phi^*(x,y)dxdy \\
&\leq \iint \max(z/2, 1)\left[ |\|x - C\| - \|y - C\|| \cdot (\|x - C\|^{z-1} + \|y - C\|^{z-1}) \right]\phi^*(x,y)dxdy \\
&\leq 2\max(z/2, 1) \iint |\|x - C\| - \|y - C\|| \phi^*(x,y)dxdy \ ,
\end{aligned}$$

where in the second equality we used the fact that $\phi^*$ is a coupling, the third inequality follows from Proposition 6, and the last inequality is due to the fact that $P$ and $Q$ have their support in the ball of radius $1/2$. For a point $y$, let $y^C$ be the closest center of $C$ to $y$. We have that $\|x - C\| - \|y - C\| \leq \|x - y^C\| - \|y - y^C\| \leq \|x - y\|$. With a similar strategy, it is also possible to show $\|y - C\| - \|x - C\| \leq \|x - y\|$. Thus, it holds:

$$2 \max(z/2, 1) \iint |\|x - C\| - \|y - C\|| \, \phi^*(x, y) dx dy$$

$$\leq 2 \max(z/2, 1) \iint (\|x - y\|) \phi^*(x, y) dx dy$$

$$= \max(z, 2) \boldsymbol{W}(P, Q) \ .$$

We can prove the same upper bound for $\mathcal{W}_{k,z}(C, P) - \mathcal{W}_{k,z}(C, Q)$, and we can conclude by observing that the inequality holds for any $C \in \mathcal{C}$. ∎

### A.3. Discrepancy Inequalities

In this subsection, we will prove inequalities involving the discrepancy, including Lemma 1 and Lemma 3. We first need the following auxilary result.

**Proposition 8** *For any* $1 \leq w \leq T$*, the following inequality holds*

$$\tau(P_T, P_T^{[w]}) \leq \max_{0 \leq t < w} \tau(P_T, P_{T-t}) \ .$$

**Proof** We use the definition of $\tau$ and $P_T^{[w]}$ and write

$$\tau(P_T, P_T^{[w]}) = \sup_{C \in \mathcal{C}} \left| \mathcal{W}_{k,z}(C, P_T) - \mathcal{W}_{k,z}(C, P_T^{[w]}) \right| = \sup_{c \in \mathcal{C}} \left| \mathcal{W}_{k,z}(C, P_T) - \frac{1}{w} \sum_{t=T-w+1}^{T} W(C, P_t) \right|$$

$$= \sup_{C \in \mathcal{C}} \left| \frac{1}{w} \sum_{t=T-w+1}^{T} [\mathcal{W}_{k,z}(C, P_T) - \mathcal{W}_{k,z}(C, P_t)] \right|$$

$$\leq \frac{1}{w} \sum_{t=T-w+1}^{T} \sup_{C \in \mathcal{C}} |\mathcal{W}_{k,z}(C, P_T) - \mathcal{W}_{k,z}(C, P_t)| = \frac{1}{w} \sum_{0 \leq t < w} \tau(P_T, P_{T-t}) \ ,$$

where the inequality is due to the triangle inequality. We can conclude that

$$\tau(P_T, P_T^{[w]}) \leq \frac{1}{w} \sum_{0 \leq t < w} \tau(P_T, P_{T-t}) \leq \frac{1}{w} \sum_{0 \leq t < w} \sup_{t' < w} \tau(P_T, P_{T-t'}) = \sup_{t' < w} \tau(P_T, P_{T-t'}) \ .$$

∎

**Proof** [Lemma 1] We apply the triangle inequality and obtain that

$$\tau(P_T, \mathbb{P}_T^{[w]}) \leq \tau(P_T^{[w]}, \mathbb{P}_T^{[w]}) + \tau(P_T^{[w]}, P_T) \ .$$

We conclude the proof using Proposition 8 ■

**Proof** [Lemma 3] The following chain of inequalities holds:

$$\mathcal{W}_{k,z}(\tilde{C}_T^{[w]}, P_T) - \beta \mathcal{W}_{k,z}^*(P_T)$$
$$= \mathcal{W}_{k,z}(\tilde{C}_T^{[w]}, P_T) - \mathcal{W}_{k,z}(\tilde{C}_T^{[w]}, \mathbb{P}_T^{[w]}) + \mathcal{W}_{k,z}(\tilde{C}_T^{[w]}, \mathbb{P}_T^{[w]}) - \beta \mathcal{W}_{k,z}^*(P_T)$$
$$\leq |\mathcal{W}_{k,z}(\tilde{C}_T^{[w]}, P_T) - \mathcal{W}_{k,z}(\tilde{C}_T^{[w]}, \mathbb{P}_T^{[w]})| + \beta \mathcal{W}_{k,z}(\hat{C}_T^{[w]}, \mathbb{P}_T^{[w]}) - \beta \mathcal{W}_{k,z}^*(P_T). \quad (11)$$

Let $C^*$ be an optimal solution with respect to $P_T$. We have that $\mathcal{W}_{k,z}^*(P_T) = \mathcal{W}_{k,z}(C^*, P_T)$ and $\mathcal{W}_{k,z}(\hat{C}_T^{[w]}, \mathbb{P}_T^{[w]}) \leq \mathcal{W}_{k,z}(C^*, \mathbb{P}_T^{[w]})$. Therefore, we can upper bound (11) as

$$\mathcal{W}_{k,z}(\tilde{C}_T^{[w]}, P_T) - \beta \mathcal{W}_{k,z}^*(P_T)$$
$$\leq |\mathcal{W}_{k,z}(\tilde{C}_T^{[w]}, P_T) - \mathcal{W}_{k,z}(\tilde{C}_T^{[w]}, \mathbb{P}_T^{[w]})| + \beta |\mathcal{W}_{k,z}(C^*, \mathbb{P}_T^{[w]}) - \beta \mathcal{W}_{k,z}(C^*, P_T)|$$
$$\leq (\beta + 1) \cdot \tau(P_T, \mathbb{P}_T^{[w]}) \ ,$$

where in the last inequality we used the definition of the metric $\tau$. The statement follows by an application of Lemma 1. ■

## Appendix B. Upper Bound (Theorem 2)

In this section, we provide the full analysis of the algorithm that attains the guarantee of Theorem 2.

As a preliminary step, we formally instantiate and prove the upper bound to the statistical error. We remark that this error is due to the fact that we use the empirical distribution $\mathbb{P}_T^{[w]}$ to approximate the cost of a solution with respect to the underlying distribution $P_T^{[w]}$. This error has been extensively studied for the special case of $(k, 2)$-approximation in the case of independent samples from the same distribution (e.g., Biau et al., 2008; Maurer, 2016; Li and Liu, 2021), and recent work generalize those results for the $(k, z)$-approximation problem for constant $z$ (Bucarelli et al., 2023). We provide an upper bound to the statistical error with an explicit dependency on $z$.

**Lemma 9** *Consider the $(k, z)$-approximation problem, where $z \geq 1$. Let $\delta \in (0, 1)$ and $1 \leq w \leq T$. There exists a constant $c > 0$ such that with probability at least $1 - \delta$, it holds that:*

$$\tau(P_T^{[w]}, \mathbb{P}_T^{[w]}) \leq c \left[ \sqrt{\frac{z^2 k}{w} \log^4(w)} + \sqrt{\frac{\ln(1/\delta)}{w}} \ \right].$$

The proof of this result is provided in Subsection B.1. The guarantee of our algorithm is conditioned on the event that we obtain a good estimation of the costs with respect to $P_T^{[w_j]}$ by using the empirical distribution $\mathbb{P}_T^{[w_j]}$, for all $j \geq 0$ (we remind that $w_j = 2^j$). This is formalized in the following corollary, which follows by carefully taking a union bound over the events of Lemma 9.

**Corollary 10** *Let $\delta \in (0, 1)$. There exists a constant $c > 0$ such that with probability at least $1 - \delta$, for all $j \geq 0$, it holds that:*

$$\tau(P_T^{[w_j]}, \mathbb{P}_T^{[w_j]}) \leq c \left[ \sqrt{\frac{z^2 k}{w_j} \log^4(w_j)} + \sqrt{\frac{\ln(1/\delta) + \log(\log w_j + 1)}{w_j}} \ \right] \ .$$

**Proof** Let $\delta_j = (6/\pi^2)\delta/(j+1)^2$. Lemma 9 implies that for the window size $w_j$, we have that with probability at least $1 - \delta_j$, it holds

$$\tau(P_T^{[w_j]}, \mathbb{P}_T^{[w_j]}) \leq c \left[ \sqrt{\frac{z^2 k}{w_j} \log^4(w_j)} + \sqrt{\frac{\ln(1/\delta_j)}{w_j}} \right]$$

$$= c \left[ \sqrt{\frac{z^2 k}{w_j} \log^4(w_j)} + \sqrt{\frac{\ln(\pi^2/(6\delta)) + 2\log(\log w_j + 1)}{w_j}} \right] .$$

Note that $\sum_{j \geq 0}(j+1)^{-2} \leq \pi^2/6$. By taking a union bound, the above event holds for all $j \geq 0$ with probability at least $1 - \sum_{j \geq 0} \delta_j \geq 1 - \delta$. ∎

The function $G : \mathbb{N} \mapsto \mathbb{R}$ is defined as the right-hand side of Corollary 10 (compare to Equation (5)):

$$G(w) \doteq c \left[ \sqrt{\frac{z^2 k}{w} \log^4(w)} + \sqrt{\frac{\ln(1/\delta) + \log(\log w + 1)}{w}} \right] .$$

We remind that $\tilde{C}_T^{[w_j]}$ is defined as the solution that minimizes the cost computed with respect to the empirical distribution $\mathbb{P}_T^{[w_j]}$ among all the approximate solutions computed by $\mathcal{A}$ on windows of size $w_i$ with $0 \leq i \leq j$,

$$\tilde{C}_T^{[w_j]} = \text{argmin} \left\{ \mathcal{W}_{k,z}\left(C, \mathbb{P}_T^{[w_j]}\right) : C \in \{\mathcal{A}(\mathbb{P}_T^{[w_i]})\}_{0 \leq i \leq j} \right\}. \tag{12}$$

To prove Theorem 2 we must show that the solution returned by Algorithm 1 exhibits the quality stated in the theorem. As sketched in Section 5, the argument revolves around two lemmas. The first lemma shows that if at iteration $j$ the if condition is false for all $i = 0, \ldots, j - 1$, then the guarantee provided by the solution $\tilde{C}_T^{[w_j]}$ is up to constants as good as the one provided by any of the previous solutions (Lemma 11). If that is the case, we can keep iterating.

**Lemma 11** *If the following condition holds for all $i$ such that $0 \leq i \leq j$:*

$$\mathcal{W}_{k,z}(\tilde{C}_T^{[w_j]}, \mathbb{P}_T^{[w_i]}) - \mathcal{W}_{k,z}(\tilde{C}_T^{[w_i]}, \mathbb{P}_T^{[w_i]}) \leq 6G(w_j) + 2G(w_i) ,$$

*then we have that*

$$\mathcal{W}_{k,z}(\tilde{C}_T^{[w_j]}, P_T) \leq \beta \cdot \mathcal{W}_{k,z}^*(P_T) + \min_{0 \leq i \leq j} \left[ (\beta + 1) \cdot \tau(P_T, \mathbb{P}_T^{[w_i]}) + 6G(w_j) + 2G(w_i) \right] .$$

**Proof** Fix an integer $i$ such that $0 \leq i \leq j$. We have that

$$\mathcal{W}_{k,z}(\tilde{C}_T^{[w_j]}, P_T) - \beta \cdot \mathcal{W}_{k,z}^*(P_T)$$
$$= \mathcal{W}_{k,z}(\tilde{C}_T^{[w_j]}, P_T) - \beta \cdot \mathcal{W}_{k,z}^*(P_T) + \mathcal{W}_{k,z}(\tilde{C}_T^{[w_j]}, \mathbb{P}_T^{[w_i]}) - \mathcal{W}_{k,z}(\tilde{C}_T^{[w_j]}, \mathbb{P}_T^{[w_i]}) . \tag{13}$$

Now observe that

$$\mathcal{W}_{k,z}(\tilde{C}_T^{[w_j]}, \mathbb{P}_T^{[w_i]}) \leq \mathcal{W}_{k,z}(\tilde{C}_T^{[w_i]}, \mathbb{P}_T^{[w_i]}) + 6G(w_j) + 2G(w_i)$$
$$\leq \beta \cdot \mathcal{W}_{k,z}^*(\mathbb{P}_T^{[w_i]}) + 6G(w_j) + 2G(w_i)$$
$$\leq \beta \cdot \mathcal{W}_{k,z}(C^*, \mathbb{P}_T^{[w_i]}) + 6G(w_j) + 2G(w_i) ,$$

where $C^*$ is a solution such that $\mathcal{W}^*_{k,z}(P_T) = \mathcal{W}_{k,z}(C^*, P_T)$. By plugging the above inequality into (13), we obtain that

$$
\begin{aligned}
&\mathcal{W}_{k,z}(\tilde{C}_T^{[w_j]}, P_T) - \beta \cdot \mathcal{W}^*_{k,z}(P_T) \\
\leq\ & \left( \mathcal{W}_{k,z}(\tilde{C}_T^{[w_j]}, P_T) - \mathcal{W}_{k,z}(\tilde{C}_T^{[w_j]}, \mathbb{P}_T^{[w_i]}) \right) \\
&+ \beta \left( \mathcal{W}_{k,z}(C^*, \mathbb{P}_T^{[w_i]}) - \mathcal{W}_{k,z}(C^*, P_T) \right) + 6G(w_j) + 2G(w_i) \\
\leq\ & (\beta + 1) \cdot \tau(\mathbb{P}_T^{[w_i]}, P_T) + 6G(w_j) + 2G(w_i) \ .
\end{aligned}
$$

∎

The second proposition shows that if at iteration $j$ there exists $i < j$ such that the `if` condition is true, then a sizeable distribution drift occurred from $P_{T-w_j+1}$ to $P_T$. When this happens, we can interrupt the iteration loop of our algorithm (Lemma 12).

**Lemma 12** *Assume that the event of Corollary 10 holds. If there exists $i$ and $j$, with $0 \leq i < j$, such that*

$$
\mathcal{W}_{k,z}(\tilde{C}_T^{[w_j]}, \mathbb{P}_T^{[w_i]}) - \mathcal{W}_{k,z}(\tilde{C}_T^{[w_i]}, \mathbb{P}_T^{[w_i]}) \geq 6G(w_j) + 2G(w_i)
$$

*then $\max_{0 \leq t < w_j} \tau(P_T, P_{T-t}) \geq G(w_j)$.*

**Proof** The proof of this proposition relies on the following inequality

$$
\tau(P_T^{[w_j]}, P_T^{[w_i]}) \leq \tau(P_T, P_T^{[w_j]}) + \tau(P_T, P_T^{[w_i]}) \leq 2 \sup_{t < w_j} \tau(P_T, P_{T-t}) \ , \tag{14}
$$

where the last step is due to Proposition 8.

Let $i$ and $j$ be the indexes defined in the statement of the proposition. We divide the analysis into two cases: $(a)$ when inequality

$$
W(\tilde{C}_T^{[w_j]}, \mathbb{P}_T^{[w_i]}) - \mathcal{W}_{k,z}(\tilde{C}_T^{[w_j]}, \mathbb{P}_T^{[w_j]}) \geq 3G(w_j) + G(w_i) \tag{15}
$$

holds, or $(b)$ when inequality (15) does not hold.

Case $(a)$. We are assuming that (15) holds. In this case, we have that

$$
\begin{aligned}
\mathcal{W}_{k,z}(\tilde{C}_T^{[w_j]}, \mathbb{P}_T^{[w_i]}) - \mathcal{W}_{k,z}(\tilde{C}_T^{[w_j]}, \mathbb{P}_T^{[w_j]}) &\leq \tau(\mathbb{P}_T^{[w_i]}, \mathbb{P}_T^{[w_j]}) \\
&\leq \tau(\mathbb{P}_T^{[w_i]}, P_T^{[w_i]}) + \tau(P_T^{[w_i]}, P_T^{[w_j]}) + \tau(P_T^{[w_j]}, \mathbb{P}_T^{[w_j]}) \\
&\leq \tau(P_T^{[w_i]}, P_T^{[w_j]}) + G(w_i) + G(w_j) \ .
\end{aligned}
$$

The second inequality is obtained by using the triangle inequality, and the third inequality is due to Corollary 10. By combining the above inequality with (14), we obtain that $\tau(P_T^{[w_i]}, P_T^{[w_j]}) \geq G(w_j)$.

Case $(b)$. Inequality (15) does not hold. In this case, we have that:

$$\mathcal{W}_{k,z}(\tilde{C}_T^{[w_j]}, \mathbb{P}_T^{[w_i]}) - \mathcal{W}_{k,z}(\tilde{C}_T^{[w_i]}, \mathbb{P}_T^{[w_i]})$$
$$\leq \mathcal{W}_{k,z}(\tilde{C}_T^{[w_j]}, \mathbb{P}_T^{[w_j]}) - \mathcal{W}_{k,z}(\tilde{C}_T^{[w_i]}, \mathbb{P}_T^{[w_i]}) + G(w_i) + 3G(w_j)$$
$$\leq \mathcal{W}_{k,z}(\tilde{C}_T^{[w_i]}, \mathbb{P}_T^{[w_j]}) - \mathcal{W}_{k,z}(\tilde{C}_T^{[w_i]}, \mathbb{P}_T^{[w_i]}) + G(w_i) + 3G(w_j)$$
$$\leq \tau(\mathbb{P}_T^{[w_i]}, \mathbb{P}_T^{[w_j]}) + G(w_i) + 3G(w_j)$$
$$\leq \tau(\mathbb{P}_T^{[w_i]}, P_T^{[w_i]}) + \tau(P_T^{[w_i]}, P_T^{[w_j]}) + \tau(P_T^{[w_j]}, \mathbb{P}_T^{[w_j]}) \quad + G(w_i) + 3G(w_j)$$
$$\leq \tau(P_T^{[w_i]}, P_T^{[w_j]}) + 2G(w_i) + 4G(w_j) \ .$$

The first inequality is due to (15), the second inequality is obtained by using the definition of $\tilde{C}_T^{[w_j]}$ (12), and the last inequality is due to Corollary 10. By using the assumption of the proposition, we have that

$$\tau(P_T^{[w_i]}, P_T^{[w_j]}) \geq 6G(w_j) + 2G(w_i) - 4G(w_j) - 2G(w_i) = 2G(w_j) \ .$$

We can conclude the proof by using inequality (14). ∎

The above lemma provides a lower bound to the drift error when the stopping condition is verified. This is the crucial technical result that allows the algorithm to lower bound the drift without explicitly computing it. The proof of the theorem is constructed with those two lemmas.

**Proof** [Theorem 2] Consider the function $\Phi(w)$ defined as

$$\Phi(w) = \beta \left[ G(w) + \sup_{t<w} \tau(P_T, P_{T-t}) \right] \ .$$

Let $w^* = \operatorname{argmin}_{1 \leq w \leq T} \Phi(w)$ be the optimal window size according to $\Phi$. The algorithm returned $\tilde{C}^{[w_j]}$ for some $j$ such that $0 \leq j \leq \lfloor \log_2 T \rfloor$. To prove the theorem, it is sufficient to show that

$$\frac{\mathcal{W}_{k,z}(\tilde{C}^{[w_j]}) - \beta \cdot \mathcal{W}_{k,z}^*(P_T)}{\Phi(w^*)} = O(1) \ .$$

We assume that the event of Corollary 10 holds, otherwise we say that our algorithm fails (with probability $\leq \delta$). Let $\ell$ be the largest integer such that $2^\ell \leq w^*$. We distinguish two cases: $\ell \leq j$ and $\ell > j$.

(Case $\ell \leq j$). Since the algorithm returned the value $j$, we have that Lemma 11 holds for $j$. As $\ell \leq j$, we obtain

$$\frac{\mathcal{W}_{k,z}(\tilde{C}^{[w_j]}) - \beta \cdot \mathcal{W}_{k,z}^*(P_T)}{\Phi(w^*)}$$
$$\leq \frac{(\beta+1) \cdot \tau(P_t, \mathbb{P}_T^{[w_\ell]}) + 6G(w_j) + 2G(w_\ell)}{\Phi(w^*)}$$
$$\leq \frac{(\beta+1) G(w_\ell) + (\beta+1) \sup_{t<w_\ell} \tau(P_T, P_{T-t}) + 6G(w_j) + 2G(w_\ell)}{\beta G(w^*) + \beta \sup_{t<w^*} \tau(P_T, P_{T-t})}$$
$$\leq \left(1 + \frac{3}{\beta}\right) \frac{G(w_\ell)}{G(w^*)} + \left(1 + \frac{1}{\beta}\right) \frac{\sup_{t<w_\ell} \tau(P_T, P_{T-t})}{\sup_{t<w^*} \tau(P_T, P_{T-t})} + \frac{6}{\beta} \frac{G(w_j)}{G(w^*)}$$
$$= O(1) \ ,$$

where the second inequality is due to Lemma 1.

(Case $\ell > j$). Since the algorithm terminated with $j < \ell$, it means that there exists a value of $i$ such that the iteration of the algorithm terminated with indexes $(j + 1, i)$. Due to Lemma 12, this implies that

$$\sup_{t < w^*} \tau(P_T, P_{T-t}) \geq \sup_{t < w_{j+1}} \tau(P_T, P_{T-t}) \geq G(w_{j+1}) \ . \tag{16}$$

We have that

$$\frac{\mathcal{W}_{k,z}(\tilde{C}^{[w_j]}) - \beta \cdot \mathcal{W}_{k,z}^*(P_T)}{\Phi(w^*)}$$
$$\leq \frac{(\beta + 1)\, G(w_j) + (\beta + 1) \sup_{t < w_j} \tau(P_T, P_{T-t})}{\beta G(w^*) + \beta \sup_{t < w^*} \tau(P_T, P_{T-t})}$$
$$\leq \left(1 + \frac{1}{\beta}\right) \frac{G(w_j)}{\sup_{t < w^*} \tau(P_T, P_{T-t})} + \left(1 + \frac{1}{\beta}\right) \frac{\sup_{t < w_j} \tau(P_T, P_{T-t})}{\sup_{t < w^*} \tau(P_T, P_{T-t})}$$
$$\leq \left(1 + \frac{1}{\beta}\right) \frac{G(w_j)}{G(w_{j+1})} + \left(1 + \frac{1}{\beta}\right) = O(1) \ .$$

The first inequality follows from Lemma 1, and the third inequality is due to (16). ∎

### B.1. Upper bound on the Statistical Error (Lemma 9)

In this subsection, we prove the upper bound on the statistical error of Lemma 9 for the $(k, z)$-approximation problem. The statistical error of the $(k, z)$-approximation problem has already been studied by Bucarelli et al. (2023) for arbitrary $k$ and constant $z$. In particular, we will use their main result for the case $1 \leq z < 2$.

**Lemma 13 (Bucarelli et al. (2023))** *Let $1 \leq z < 2$, $w \in \{1, \ldots, T\}$, and $\delta \in (0, 1)$. There exists a constant $c$ such that with probability at least $1 - \delta$, it holds that:*

$$\tau(P_T^{[w]}, \mathbb{P}_T^{[w]}) \leq c \cdot \left( \sqrt{\frac{k \log^4(w)}{w}} + \sqrt{\frac{\log(1/\delta)}{w}} \right) \ .$$

**Proof** This result is a straightforward adaptation of Theorem 5.1 in Bucarelli et al. (2023) to our setting. ∎

Lemma 13 shows that Lemma 9 is true for $1 \leq z < 2$. In the remaining of this subsection, we will focus on the case $z \geq 2$. Compared to the previous work, our goal is to provide a characterization of the statistical error with an explicit dependency on $z$. To obtain this result, we extend the argument by Liu (2021) based on the Rademacher complexity for $(k, 2)$-approximation problem to an arbitrary $z \geq 2$.

Consider the family of functions $\mathcal{F} : \mathbb{R}^d \mapsto \mathbb{R}$ defined as:

$$\mathcal{F} = \{x \mapsto \|x - C\|^z : C \in \mathcal{C}\} \ .$$

The empirical Rademacher complexity of the family $\mathcal{F}$ (Shalev-Shwartz and Ben-David, 2014) over the last $w$ steps is defined as:

$$\hat{\mathcal{R}}_w(\mathcal{F}) \doteq \mathbb{E}_\sigma \sup_{f \in \mathcal{F}} \left( \sum_{t=T-w+1}^{T} \sigma_t f(X_t) \right) \;,$$

where $\sigma_1, \ldots, \sigma_T$ are independent Rademacher random variables, i.e., $\sigma_t = \pm 1$ with probability $1/2$. The Rademacher complexity of $\mathcal{F}$ over the previous $w$ steps is $\mathcal{R}_w(\mathcal{F}) \doteq \mathbb{E}_X \hat{\mathcal{R}}_w(\mathcal{F})$. The following result provides an upper bound to the empirical Rademacher complexity, and it will be proven at the end of this subsection.

**Lemma 14** *Let $z \geq 2$, and let $\mathcal{X} = \{x : \|x\|^2 \leq 1/2\}$. There exists a constant $c > 0$ such that for any set of samples $\{X_{T-w+1}, \ldots, X_T\} \in \mathcal{X}^w$, we have that:*

$$\hat{\mathcal{R}}_w(\mathcal{F}) \leq c \cdot z \sqrt{kw \log^4(w)} \;.$$

The above upper bound to the empirical Rademacher complexity can be used to prove an upper bound to the statistical error through classical learning theory techniques (Mohri et al., 2018).

**Proof** [Lemma 9] If $1 \leq z < 2$, the result immediately follows from Lemma 13. Let $z \geq 2$. We can observe that $\|f\|_\infty \leq 1$ for any $f \in \mathcal{F}$ since all the points are contained in the ball of radius $1/2$ centered in the origin. We can use McDiarmid's inequality, and show that with probability at least $1 - \delta/2$ it holds that:

$$\frac{1}{w}\mathcal{R}_w(\mathcal{F}) \leq \frac{1}{w}\hat{\mathcal{R}}_w(\mathcal{F}) + O\left( \sqrt{\frac{\log(1/\delta)}{w}} \right) \;. \tag{17}$$

We use Lemma 14 to upper bound the empirical Rademacher complexity. Thus, we have that:

$$\frac{1}{w}\mathcal{R}_w(\mathcal{F}) = O\left( z\sqrt{\frac{k}{w}\log^4(w)} + \sqrt{\frac{\ln(1/\delta)}{w}} \right) \;. \tag{18}$$

Let $A$ be the random variable of the statement of the lemma

$$A = \tau(P_T^{[w]}, \mathbb{P}_T^{[w]}) \;. \tag{19}$$

By a standard symmetrization argument (Vershynin, 2018), we have that

$$\mathbb{E}\,A \leq (2/w)\mathcal{R}_w(\mathcal{F}) \;. \tag{20}$$

We can use McDiarmid's inequality again and show that with probability at least $1 - \delta/2$, it holds:

$$A \leq \mathbb{E}\,A + O\left( \sqrt{\frac{\ln(1/\delta)}{w}} \right) \;. \tag{21}$$

By combining (18), (20) and (21), and taking a union bound, we obtain the final statement. ∎

The remaining of this subsection is dedicated to the proof of Lemma 14. The first step is the application of the $\ell_\infty$ vector contraction for the Rademacher complexity (Foster and Rakhlin, 2019).

**Lemma 15** *Let $\mathcal{X} = \{x : \|x\|^2 \leq 1/2\}$. There exists a constant $c > 0$ such that for any set of samples $\{X_{T-w+1}, \ldots, X_T\} \in \mathcal{X}^w$, we have that:*

$$\hat{\mathcal{R}}_w(\mathcal{F}) \leq c\sqrt{k}\tilde{\mathcal{R}}_w(\overline{\mathcal{F}})\log^2\left(\frac{w}{\tilde{\mathcal{R}}_w(\overline{\mathcal{F}})}\right) \;,$$

*where $\overline{\mathcal{F}} = \{x \mapsto \|x - c\|^z : c \in \mathcal{X}\}$, and $\tilde{\mathcal{R}}_w(\overline{\mathcal{F}}) = \sup_{\{X_{T-w+1}, \ldots, X_T\} \in \mathcal{X}^w} \hat{\mathcal{R}}_w(\overline{\mathcal{F}})$.*

**Proof** It is a straightforward adaptation of the proof of Lemma 1 in Liu (2021) to our setting. ∎

Lemma 15 shows that in order to upper bound the empirical Rademacher complexity of $\mathcal{F}$, it is sufficient to evaluate the worst-case empirical Rademacher complexity $\tilde{\mathcal{R}}_w(\overline{\mathcal{F}})$, where $\overline{\mathcal{F}}$ is a family of functions that is restricted to consider a single center. To this end, we show both an upper bound (Proposition 16) and a lower bound (Proposition 17) to $\tilde{\mathcal{R}}_w(\overline{\mathcal{F}})$.

**Proposition 16 (Upper bound to worst-case empirical Rademacher complexity)** *The following inequality holds:*

$$\tilde{\mathcal{R}}_w(\overline{\mathcal{F}}) \leq \frac{3z}{4}\sqrt{w} \;.$$

**Proof** Let $\mathcal{X} = \{x \in \mathbb{R}^d : \|x\|^2 \leq 1/2\}$. Consider an arbitrary input $\{X_{T-w+1}, \ldots, X_T\} \in \mathcal{X}^w$. We have that:

$$\hat{\mathcal{R}}_w(\overline{\mathcal{F}}) = \mathbb{E}_\sigma \sup_{f \in \overline{\mathcal{F}}} \left(\sum_{t=T-w+1}^{T} \sigma_t f(X_t)\right) = \mathbb{E}_\sigma \sup_{c \in \mathcal{X}} \left(\sum_{t=T-w+1}^{T} \sigma_t \|X_t - c\|^z\right) \;.$$

For $t \in \{T - w + 1, \ldots, T\}$, let $\psi_t : \mathcal{X} \mapsto \mathbb{R}$ be defined as $\psi_t(c) \doteq \|X_t - c\|^2$, and let $h : \mathbb{R} \mapsto \mathbb{R}$ be the power function $h(a) = a^{z/2}$. It is possible to observe that $\|X_t - c\|^z = h(\psi_t(c))$. Note that $0 \leq \psi_t(c) \leq 1$ for any $t$ and $c \in \mathcal{X}$, and $h$ is $(z/4)$-Lipschitz for $a \in [0, 1]$. By using the Lipschitz contraction inequality for Rademacher complexity (Ledoux and Talagrand, 2013), it is possible to show that

$$\hat{\mathcal{R}}_w(\overline{\mathcal{F}}) = \mathbb{E}_\sigma \sup_{c \in \mathcal{X}} \sum_{t=T-w+1}^{T} \sigma_t h(\psi_t(c)) \leq \frac{z}{4} \mathbb{E}_\sigma \sup_{c \in \mathcal{X}} \sum_{t=T-w+1}^{T} \sigma_t \|X_t - c\|^2 \;. \tag{22}$$

We upper bound the expectation of the right-hand side of (22) as follows:

$$\mathbb{E}_\sigma \sup_{c \in \mathcal{X}} \sum_{t=T-w+1}^{T} \sigma_t \|X_t - c\|^2 = \mathbb{E}_\sigma \sup_{c \in \mathcal{X}} \sum_{t=T-w+1}^{T} \sigma_t \left(\|X_t\|^2 + \|c\|^2 - 2\langle X_t, c\rangle\right)$$

$$\leq \mathbb{E}_\sigma \sup_{c \in \mathcal{X}} \sum_{t=T-w+1}^{T} \sigma_t \|c\|^2 + 2 \mathbb{E}_\sigma \sup_{c \in \mathcal{X}} \sum_{t=T-w+1}^{T} \sigma_t \langle X_t, c\rangle \;. \tag{23}$$

We have the following upper bounds:

$$\mathbb{E}_\sigma \sup_{c \in \mathcal{X}} \sum_{t=T-w+1}^{T} \sigma_t \|c\|^2 \leq \mathbb{E}_\sigma \sup_{c \in \mathcal{C}} \|c\|^2 \left|\sum_{t=T-w+1}^{T} \sigma_t\right| \leq \sqrt{w} \;, \tag{24}$$

where the last step is due to the Khintchine inequality (Garling, 2007). Also, it holds:

$$2\,\mathbb{E}\sup_{\sigma}\sup_{c\in\mathcal{X}}\sum_{t=T-w+1}^{T}\sigma_t\langle X_t,c\rangle = 2\,\mathbb{E}\sup_{\sigma}\sup_{c\in\mathcal{X}}\left\langle\sum_{t=T-w+1}^{T}\sigma_t X_t,c\right\rangle \leq 2\,\mathbb{E}_{\sigma}\left\|\sum_{t=T-w+1}^{T}\sigma_t X_t\right\|$$

$$\leq 2\sqrt{\mathbb{E}_{\sigma}\left\|\sum_{t=T-w+1}^{T}\sigma_t X_t\right\|^2} = 2\sqrt{\sum_{t,t'}\langle X_t,X_{t'}\rangle\,\mathbb{E}_{\sigma}\sigma_t\sigma_{t'}}$$

$$\leq 2\sqrt{w}\ , \tag{25}$$

where the first inequality is due to Jensen's inequality. By plugging (24) and (25) into the right-hand side of (23), we finally obtain that:

$$\hat{\mathcal{R}}_w(\overline{\mathcal{F}}) \leq \frac{3z}{4}\sqrt{w}\ . \tag{26}$$

The statement follows since inequality (26) holds for any choice of $\{X_{T-w+1},\ldots,X_T\}\in\mathcal{X}^w$. ∎

**Proposition 17 (Lower bound to worst-case empirical Rademacher complexity)** *The following inequality holds:*

$$\tilde{\mathcal{R}}_w(\overline{\mathcal{F}}) \geq 2^{-3/2}\sqrt{w}\ .$$

**Proof** Without loss of generality, assume that $w$ is even. Let $e_1$ be a canonical standard vector. We set half of the $w$ points to be equal to $e_1$, and the other half to be equal to $-e_1$, specifically $X_{T-w+1}=\ldots=X_{T-w/2}=e_1/2$ and $X_{T-w/2+1}=\ldots=X_T=-e_1/2$. We have that:

$$\tilde{\mathcal{R}}_w(\overline{\mathcal{F}}) \geq \mathbb{E}_{\sigma}\sup_{c\in\mathcal{C}}\left(\sum_{t=T-w+1}^{T-w/2}\sigma_t\|c-e_1\|^z + \sum_{t=T-w/2+1}^{T}\sigma_t\|c+e_2\|^z\right)$$

$$\geq \mathbb{E}_{\sigma}\max_{c\in\{e_1,-e_1\}}\left(\sum_{t=T-w+1}^{T-w/2}\sigma_t\|c-e_1\|^z + \sum_{t=T-w/2+1}^{T}\sigma_t\|c+e_2\|^z\right)$$

$$= \mathbb{E}_{\sigma}\max\left(\sum_{t=T-w+1}^{T-w/2}\sigma_t, \sum_{t=T-w/2+1}^{T}\sigma_t\right)\ .$$

Let $A=\sum_{t=T-w+1}^{T-w/2}\sigma_t$ and $B=\sum_{t=T-w/2+1}^{T}\sigma_t$. Note that $2\max(a,b)=a+b+|a-b|$ for any $a,b\in\mathbb{R}$. Thus, we have that:

$$\tilde{\mathcal{R}}_w(\overline{\mathcal{F}}) \geq \mathbb{E}_{\sigma}\max(A,B) = \frac{1}{2}\mathbb{E}_{\sigma}\left|\sum_{t=T-w+1}^{T}\sigma_t\right| \geq 2^{-3/2}\sqrt{w}$$

where the last step is obtained by an application of Khintchine inequality (Garling, 2007). ∎

**Proof** [Lemma 14] The proof follows as an immediate corollary of Lemma 15, Proposition 16 and Proposition 17. By Lemma 15, there exists a constant $c > 0$ such that for any set of samples $\{X_{T-w+1}, \ldots, X_T\}$, it holds that:

$$\hat{\mathcal{R}}_w(\mathcal{F}) \leq c\sqrt{k}\tilde{\mathcal{R}}_w(\overline{\mathcal{F}}) \log^2\left(\frac{w}{\tilde{\mathcal{R}}_w(\overline{\mathcal{F}})}\right) \ .$$

By using Proposition 16, we upper bound $\tilde{\mathcal{R}}_w(\overline{\mathcal{F}}) \leq \frac{3z}{4}$. Additionally, the statement of Proposition 17 implies that $\sqrt{w}/\tilde{\mathcal{R}}_w(\overline{\mathcal{F}}) \leq \sqrt{w}2^{3/2}$. Thus, we can conclude that there exists a constant $c' > 0$ such that:

$$\hat{\mathcal{R}}_w(\mathcal{F}) \leq c'z\sqrt{kw\log^4(w)} \ .$$

$\blacksquare$

## Appendix C. Lower Bound (Theorem 4)

This section is devoted to the proof of the lower bound of Theorem 4. For ease of notation, we denote with $\mathcal{B}(0, r) = \{x \in \mathbb{R}^d : \|x\| \leq r\}$ the ball centered at the origin with radius $r \geq 0$. To simplify the computations, without loss of generality, we assume that the supports of the distributions $P_1, \ldots, P_T$ lie in a ball centered at the origin of radius 1 rather than $1/2$.

Let $k \geq 3$. Without loss of generality, we assume that $k$ is divisible by 3. Let $m = k/3$ and let $z_1, \ldots, z_m$ be a $12\overline{\Delta}$-net of $\mathcal{B}(0, 1 - 4\overline{\Delta})$ for some later defined $\overline{\Delta} > 0$. Let $w = (\overline{\Delta}, 0, \ldots, 0) \in \mathbb{R}^d$. We define the sets $U_{i,-1} = \{z_i - 3w, z_i - 2w\}$ and $U_{i,1} = \{z_i + 3w, z_i + 4w\}$ for $i = 1, \ldots, m$, and we let $U = \cup_{i=1}^m (U_{i,1} \cup U_{i,-1})$. By construction, the points $U$ are contained in the unit ball $\mathcal{B}(0, 1)$. Following the volume argument of Levrard (2015), a sufficient condition on $\overline{\Delta}$ to guarantee that the points $z_1, \ldots, z_m$ exist is

$$m \leq \left(\frac{1 - 4\overline{\Delta}}{18\overline{\Delta}}\right)^d \ ,$$

and observe that $\overline{\Delta} = m^{-1/d}/18$ suffices.

Let $\delta = \Delta/(3\overline{\Delta})$ and let $r = (\frac{k}{48\delta^2})^{1/3}$. Let $\xi \in \{-1, 1\}^m$. We construct a sequence of discrete distributions $P_1^\xi, \ldots, P_T^\xi$ parameterized by $\xi$. Those distributions are defined as follows. If $t \leq T - r$, the distribution $P_t^\xi$ is uniform over the points $U$:

$$P_t^\xi(x) = 1/(4m) \ , \qquad \forall x \in U_{i,\pm 1}, \ i = 1, \ldots, m \quad .$$

For $t > T - r$, the distribution $P_t^\xi$ is drifting away from the uniform distribution with a drift structure described by $\xi$:

$$P_t^\xi(x) = \frac{1 \pm \xi_i(t - T + r)\delta}{4m} \qquad \forall x \in U_{i,\pm\xi_i}, \ i = 1, \ldots, m \quad .$$

**Proposition 18** *For any $\xi$ and $1 \le t < T$, it holds that*

$$\boldsymbol{W}(P_t^\xi, P_{t+1}^\xi) \le \Delta \ .$$

**Proof** If $t < T - r$, the Wasserstein distance is trivially equal to zero. If $T \ge T - r$, then we can transport the excess mass of $P_{t+1}^\xi$ generated due to the drift in $U_{i,\xi_i}$ to the set $U_{i,-\xi_i}$. With this transport, we obtain the following upper bound to the Wasserstein distance:

$$\boldsymbol{W}(P_t^\xi, P_{t+1}^\xi) \le \sum_{i=1}^m \frac{2\delta}{4m} 6\overline{\Delta} = 3\overline{\Delta}\delta = \Delta \ ,$$

where the last equality is due to the definition of $\delta$. ∎

We denote with $\tilde{\mathcal{C}}$ the collection of all set of $k$ centers $C \in \mathcal{C}$ such that for each $i = 1, \ldots, m$, either:

1. $C$ has centers $U_{i,1}$ and $z_i - 5w/2$ (average of the points in $U_{i,-1}$)

2. $C$ has centers $U_{i,-1}$ and $z_i + 7w/2$ (average of the points in $U_{i,1}$).

For $\xi \in \{-1, 1\}^m$, we denote with $C^\xi \in \tilde{\mathcal{C}}$ the solution such that if $\xi_i = 1$ (resp., $\xi_i = -1$), then the case (1) (resp., case (2)) above applies. Let $h(\xi, \xi')$ be the Hamming distance between two strings $\xi$ and $\xi'$. The following proposition holds.

**Proposition 19** *Let $\xi, \xi' \in \{-1, 1\}$. Then:*

$$\mathcal{W}_{k,z}(C^\xi, P_T^{\xi'}) = \mathcal{W}_{k,z}^*\left(P_T^{\xi'}\right) + h(\xi, \xi')\frac{r\delta\overline{\Delta}^2}{4m} \ .$$

*Also, for any $C \in \mathcal{C}$, there exists $\tilde{C} \in \tilde{\mathcal{C}}$ such that for all $\xi \in \{-1, 1\}^m$, it holds that*

$$\mathcal{W}_{k,z}(\tilde{C}, P_T^\xi) \le \mathcal{W}_{k,z}(C, P_T^\xi) \ .$$

**Proof** The first statement can be immediately obtained through a direct computation. To prove the second statement, we can notice that the set $\mathcal{Z} = \{z_i \pm 3w : 1 \le i \le m\}$ contains points that are distance at least $6\overline{\Delta}$ from one another. As each point in $U$ can be written as $z'$ or $z' + w$ for some $z' \in \mathcal{Z}$, Step 3 of Bartlett et al. (1998) shows that the statement is true if

$$6 \ge \sqrt{\frac{2}{1 - r\delta}} + 1 \ .$$

It is possible to verify that this is indeed the case with our choice of $r$ and $\delta$. ∎

Let $\mathcal{P}(\Delta)$ be the collection of all the sequence of joint distributions $P_1 \otimes \ldots \otimes P_T$ over $(X_1, \ldots, X_T) \subseteq \mathcal{B}(0, 1)^T$, such that $\boldsymbol{W}(P_t, P_{t+1}) \le \Delta$ for all $1 \le t < T$ (Due to Lemma 7, this also implies an upper bound to the discrepancy $\tau$.). For $\xi \in \{-1, 1\}^n$, consider the joint distribution $\boldsymbol{P}^\xi = P_1^\xi \otimes \ldots \otimes P_T^\xi$ over $(X_1, \ldots, X_T)$, and observe that $\boldsymbol{P}^\xi \in \mathcal{P}(\Delta)$ due to Proposition 18.

Let $\hat{C} \in \mathcal{C}$ be any solution that is decided based on the data $(X_1, \ldots, X_T)$. To prove the theorem, it is sufficient to show a lower bound to

$$\inf_{\hat{C}} \sup_{P_1 \otimes \ldots \otimes P_T \in \mathcal{P}(\Delta)} \mathbb{E}\left[\mathcal{W}_{k,z}(\hat{C}, P_T) - \mathcal{W}_{k,z}^*(P_T)\right] .$$

We have the following chain of inequalities.

$$
\begin{aligned}
& \inf_{\hat{C}} \sup_{P_1 \otimes \ldots \otimes P_T \in \mathcal{P}(\Delta)} \mathbb{E}\left[\mathcal{W}_{k,z}(\hat{C}, P_T) - \mathcal{W}_{k,z}^*(P_T)\right] \\
\geq & \inf_{\hat{C}} \sup_{\xi \in \{-1,1\}^m} \mathbb{E}\left[\mathcal{W}_{k,z}(\hat{C}, P_T^\xi) - \mathcal{W}_{k,z}^*(P_T^\xi)\right] \\
\geq & \inf_{\hat{\xi}} \sup_{\xi \in \{-1,1\}^m} \mathbb{E}\left[\mathcal{W}_{k,z}(C^{\hat{\xi}}, P_T^\xi) - \mathcal{W}_{k,z}^*(P_T^\xi)\right] \\
= & \frac{r\delta\overline{\Delta}^2}{4m} \inf_{\hat{\xi}} \sup_{\xi \in \{-1,1\}^m} \mathbb{E}\left[h(\hat{\xi}, \xi)\right] ,
\end{aligned}
\tag{27}
$$

where the last two steps are due to Proposition 19.

For two distributions $P$ and $Q$, the square of their Hellinger distance is defined as

$$H^2(P, Q) = \int (\sqrt{P(x)} - \sqrt{Q(x)})^2 dx .$$

When $\boldsymbol{P} = P_1 \otimes \ldots \otimes P_T$ and $\boldsymbol{Q} = Q_1 \otimes \ldots \otimes Q_T$, we have the following factorization property (Tsybakov, 2008):

$$H^2(\boldsymbol{P}, \boldsymbol{Q}) = 2\left[1 - \prod_{i=1}^T \left(1 - \frac{H^2(P_i, Q_i)}{2}\right)\right] .
\tag{28}$$

**Proposition 20** *Let $\xi, \xi' \in \{-1,1\}^m$ be two strings that differ only in a single position, i.e. $h(\xi, \xi') = 1$. Then:*

$$H^2(\boldsymbol{P}^\xi, \boldsymbol{P}^{\xi'}) \leq \frac{1}{2} .$$

**Proof** If $t \leq T - r$, then

$$H^2(P_t^\xi, P_t^{\xi'}) = 0 ,$$

else if $t > T - r$, then

$$
\begin{aligned}
H^2(P_t^\xi, P_t^{\xi'}) &= 4\left[\sqrt{\frac{1 + (t - T + r)\delta}{4m}} - \sqrt{\frac{1 - (t - T + r)\delta}{4m}}\right]^2 \\
&= \frac{1}{m}\left[\sqrt{1 + (t - T + r)\delta} - \sqrt{1 - (t - T + r)\delta}\right]^2 \\
&\leq \frac{1}{m}[2(t - T + r)\delta]^2 \\
&= \frac{4r^2\delta^2}{m} ,
\end{aligned}
$$

where the inequality is due to the fact that $\sqrt{x} - \sqrt{y} \leq x - y$ if $x > 1$ and $0 \leq y \leq x$. By using the factorization property (28), we obtain

$$H^2(\boldsymbol{P}^\xi, \boldsymbol{P}^{\xi'}) \leq 2\left[1 - \left(1 - \frac{4r^2\delta^2}{m}\right)^r\right] \leq \frac{8r^3\delta^2}{m} \ .$$

The statement follows by substituting the values of $r, \delta$ and $m$. ∎

We use Assouad's lemma as in Theorem 2.12 of Tsybakov (2008) to lower bound (27). Let $\alpha \doteq \sup_{\xi,\xi':h(\xi,\xi')=1} H^2(\boldsymbol{P}^\xi, \boldsymbol{P}^{\xi'})$. We have that if $\alpha \leq 2$, then

$$\inf_{\hat{\xi}} \sup_{\xi \in \{-1,1\}^m} \mathbb{E}\left[h(\hat{\xi}, \xi)\right] \geq \frac{m}{4}(1 - \sqrt{\alpha(1 - \alpha/4)}) \ .$$

By using Proposition 20 in the above inequality and combining it with (27), we finally have that:

$$\inf_{\hat{C}} \sup_{P_1 \otimes ... \otimes P_T \in \mathcal{P}(\Delta)} \mathbb{E}\left[\mathcal{W}_{k,z}(\hat{C}, P_T) - \mathcal{W}^*_{k,z}(P_T)\right] \geq \frac{r\delta\overline{\Delta}^2}{48} \ .$$

We conclude the proof of the Theorem by substituting the values of $r, \delta$ and $\overline{\Delta}$.

