# OpenReview forum: "Center-Based Approximation of a Drifting Distribution"
_algorithmiclearningtheory.org/ALT/2025/Conference — ALT 2025_

### Official Review · Reviewer_DX3Z · 2024-11-03

**Rating:** 7
**Confidence:** 4

**Review:**

This paper studies the problem of computing center-based approximations of distributions that change over time, a.k.a, drifting distributions.
Specifically, for a given positive integer parameter k, the goal is to output k centers that best represent the current distribution, meaning that they minimize the average, for a point sampled from the (unknown) distribution, of the z-th power of the L2-distance between the point and its closest center.
The paper formalizes this setting and provides both an algorithm and a lower bound.
The algorithm proceeds by running any given approximate clustering algorithm on a sliding window of points, where the window size is dynamically set as a function of the (unknown) drift of the underlying distribution. The paper proves a clean upper bound on the error this algorithm (Theorem 2 and Lemma 3) in terms of the approximation factor of the given clustering algorithm as well as the statistical and drift errors.
On the lower bound size, the paper shows for the k-means objective and in the regime where the approximation factor goes to 1 and the dimension goes to infinity, their upper bound is asymptotically tight.
Finally, the paper runs an empirical evaluation on a synthetic dataset.

The strengths of this paper are:

1) The algorithm is quite natural, and it is plausible that a variant of it could be used in practice.
2) The results are neat, and the paper is well-written.

The weaknesses of this paper are:

1) It’s unclear how natural the lower bound construction is.
2) It’s unclear how tight the lower bound is in general (in terms of the exponent of k and Delta).
3) It's not very clear how valuable the lower bound is in the regime as the approximation factor goes to 1, since k-means clustering is NP-hard in that case. I feel that the paper would benefit of some discussion of this.
4) The empirical evaluation is limited to a synthetic distribution. (This is unfortunate as it would have been valuable to see how well the algorithm performs on a real dataset. That said, this shouldn’t be held against the paper given the scope of ALT.)

Overall, I feel that the strengths of this submission outweigh its weaknesses, and that it would be a good addition to ALT. I support acceptance.

**Paper Award:**

No

---

> ### Author Response · Authors · 2024-11-23
> **Rebuttal**
>
> We thank the reviewer for their time in reviewing our paper and for their helpful comments.
> In the following reply, the citations appear as in our paper.
>
> **Reviewer’s Comment.** *The weaknesses of this paper are: [...]*
>
> *2. It’s unclear how tight the lower bound is in general (in terms of the exponent of k and Delta).*
>
> *3. It's not very clear how valuable the lower bound is in the regime as the approximation factor goes to 1, since k-means clustering is NP-hard in that case. I feel that the paper would benefit of some discussion of this.*
>
>
> As the reviewer correctly pointed out, the lower bound is only information-theoretic and does not account for the computational complexity of the algorithm. In particular, our lower bound shows that all algorithms — whether polynomial-time or exponential-time — exhibit an error due to the limited information a finite sample can provide about the distributions.
>
> We agree that restricting this lower bound to polynomial-time algorithms is an interesting direction for future work. However, we emphasize that even in the simpler case of i.i.d. data (no drift), the sample complexity lower bound for $(k,2)$-approximation ($k$-means) is only established with respect to any (possibly inefficient) algorithm. Deriving tighter bounds by focusing solely on polynomial-time algorithms is a challenging problem that is beyond the scope of this work.
>
> Our information-theoretic lower bound is tight, up to polylogarithmic factors, if the dimension $d$ is allowed to go to infinity (as discussed at the end of Section 3). This is not a limitation unique to our work: even for the $(k,2)$-approximation problem (k-means) with i.i.d. data (the no-drift case), there remains a gap between the lower and upper bounds despite significant work on this topic (e.g., [Biau et al., 2008; Maurer and Pontil, 2010; Maurer, 2016; Foster and Rakhlin, 2019; Li and Liu, 2021; Bucarelli et al., 2023]). In particular, the best lower bound on the error for $(k,2)$-approximation with $n$ i.i.d. samples is $\Omega(\sqrt{\frac{k^{1-4/d}}{n}})$, while the best upper bound is $O(\sqrt{\frac{k \log n}{n}})$. Note that even for i.i.d. data, the best known lower and upper bounds match, up to a polylog factor, only if we allow the dimension $d$ to go to infinity.
>
> We believe that progress in closing this gap for the $(k,2)$-approximation problem would also lead to improved bounds for our problem with drift.  We thank the reviewer for this feedback and agree to include a more detailed discussion of these points in the next version of our work.

---

### Official Review · Reviewer_HijQ · 2024-11-09
**Review for Center-Based Approximation of a Drifting Distribution**

**Rating:** 7
**Confidence:** 3

**Review:**

## Summary

**Setting**: For a distribution $P$ on $\mathbb R^d$, a finite subset $C$ of points in $\mathbb R^d$, an integer $k \geq 1$ and a real $z \geq 1$, let $\mathcal W_{k,z}(C,P) = \int \min_{c \in C} \|x-c\|^2 dP(x)$ be the quantity that measures the cost of approximating $P$ by the set $C$. Using that notation, $\min_{C \subset \mathbb R^d : |C|=k}\mathcal W_{k,z}(C,P)$ represents the best cost of approximating $P$ using $k$ points. When $P$ is a discrete distribution on some set $S$, the previous problem is also known as $(k,z)$-clustering. In the setting of this paper, there is a sequence of distributions $P_1,P_2,\ldots P_T$  (each satisfying $Pr_{X_t \sim P_T}(\| X_t\| \leq 1/2)= 1$ ) and the goal is to approximate $P_T$ by a set $C$ of $k$ points as accurately as possible in the sense of minimizing $\mathcal W_{k,z}(C,P_T)$. One obvious approach for this is to use a window (i.e., solve the $(k,z)$-clustering problem on the last $w$ samples). What is known by prior work for that (cf. Lemma 1) is that this solution incurs some extra cost which consists of two parts: a statistical error part, and a part that depends on how much the distribution shifts in consecutive steps. The first term decreases with $w$ while the second increases. A desired solution would be to use the best window size $w$ that optimizes the trade-off between the two terms. The goal of this paper is to qualitatively achieve that, despite the fact that the distribution shifts are unknown.

**Result**: The paper provides an upper and lower bound for this problem. For the upper bound, it provides an algorithm that, given the availability of another black-box algorithm for $(k,z)$-clustering up to a multiplicative $\beta$ factor, outputs a set of $k$ points whose cost $\mathcal W_{k,z}(C,P_T)$ for approximating $P_T$ is at most
$$\beta \cdot\mathcal W^*_{k,z}(C,P_T) + \beta \cdot O \left(\min_{1 \leq w \leq T} \left(\sqrt{z^2k/w} + \max_{0 \leq t <w} \epsilon_{t,w}\right)\right),$$
where the first term is the best possible cost (with knowledge of $P_T$) and the second term consists of two parts: a term $\sqrt{z^2k/w}$ that resembles statistical error and a term where the quantities $\epsilon_{t,w}$ quantify the distribution drift in the last $w$ steps (see Theorem 2 for how these $\epsilon_{t,w}$ are defined). Qualitatively, this bound captures the best trade-off discussed in the previous paragraph. The authors show a lower bound for $z=2$ that almost matches the cost from the upper bound.

**Approach**: The algorithm uses a sliding window but adaptively determines its length $w$. It starts with a small window and iteratively doubles its length until there is evidence of a distribution shift. This evidence is quantified by comparing the solutions of the black-box $(k,z)$-clustering algorithm for different window sizes. Once a large enough gap is detected, the algorithm stops and uses that window size for its output. The lower bound consists of constructing a family of sequences of distributions and reducing the problem of learning this family (where by learning we mean approximating it with $k$ points) to a binary hypothesis testing problem for which standard information-theory machinery like Le Cam's lemma can be used.

I enjoyed reading this paper and found it generally well-written and clear. Although I am not an expert in this specific area, the introduction provided a compelling narrative, with a  review of prior work and justification for the problem’s importance. The statements of the results were clear and are followed by discussion that facilitates interpretation. I have not read carefully the details (Appendix) but the proof sketches made sense. One potential weakness (or question for the authors) is that the algorithm requires all distributions to be supported within a constant-radius Euclidean ball. Is this assumption common in the literature? How natural is it, and what changes might be expected under more specific (e.g., parametric) distribution families? Including a discussion on this assumption in the paper would be useful.

**Paper Award:**

No

---

> ### Author Response · Authors · 2024-11-23
> **Rebuttal**
>
> We thank the reviewer for their close reading of our work and their positive feedback.
> In the following reply, the citations appear as in our paper.
>
> **Reviewer’s Comment**. *One potential weakness (or question for the authors) is that the algorithm requires all distributions to be supported within a constant-radius Euclidean ball. Is this assumption common in the literature? How natural is it, and what changes might be expected under more specific (e.g., parametric) distribution families? Including a discussion on this assumption in the paper would be useful.*
>
> These are great questions. We want to highlight that the assumption of the support being bounded is standard in previous work of $(k,2)$-approximation ($k$-means) of distributions (e.g.,  [Biau et al., 2008; Maurer and Pontil, 2010; Maurer, 2016; Foster and Rakhlin, 2019; Li and Liu, 2021; Bucarelli et al., 2023]). We thank the reviewer for this feedback, and we will add a discussion on this assumption.
>
> The current techniques that provide the best upper bound to the statistical error for the $(k,2)$-approximation problem all use this boundness assumption (Foster and Rakhlin, 2019,  Li and Liu, 2021; Bucarelli et al., 2023). The boundness assumption on the support of the distribution is used to guarantee that each cost is bounded. In learning theory, a boundness assumption for the cost is common to characterize the sample complexity of learning problems ([1] gives an example for the fundamental problem of linear regression).
>
> There is no evidence that this assumption is necessary. Any assumption on the distributions that can guarantee a bounded statistical error with high probability would suffice. It is possible that other assumptions (possibly parametric) that can control the tail of the random variable $|| x - C||^z$ for a random $x \sim P$ would also work (e.g., $P$ being subgaussian), although this would require developing new techniques to upper bound the statistical error, which may be non-trivial.
>
> [1]: Shamir, Ohad. "The sample complexity of learning linear predictors with the squared loss." J. Mach. Learn. Res. 16 (2015): 3475-3486.

---

### Official Review · Reviewer_uspc · 2024-11-11
**Review report**

**Rating:** 3
**Confidence:** 3

**Review:**

This paper explores a center-based approximation approach for handling drifting distributions. Given a pre-defined k, the authors approximate recent observations in a sequence using k-center clusters. This approach is applied to a two-dimensional sequence of Gaussian mixtures with varying means, assuming independence between the dimensions and equal variance.

However, the motivation behind using a center-based approximation for a drifting distribution is unclear. For such sequences, a changing slope model would be a better fit. Changing slope models typically struggle with high-dimensional data sequences due to the complexity of parameter estimation, particularly for the covariance matrix. Yet, in this study, which focuses on two-dimensional data with a scaled identity matrix as the covariance matrix, these challenges do not arise, making the changing slope model an ideal fit.

Alternatively, even a change point model would be preferable to the k-center approximation. Unlike the k-center approach, a change-point model accounts for temporal information, clustering observations that are temporally close – an essential aspect of drifting distributions.

Furthermore, the reliance on a pre-defined k renders the model somewhat ad hoc. In contrast, both the changing slope and change-point models provide data-driven methods to determine the number of change points, enhancing flexibility and adaptability to the data.

**Paper Award:**

No

---

> ### Author Response · Authors · 2024-11-23
> **Rebuttal**
>
> We thank the reviewer for the time spent reviewing our work and for their feedback.
>
> **Reviewer’s Comment**. *However, the motivation behind using a center-based approximation for a drifting distribution is unclear. For such sequences, a changing slope model would be a better fit. …. Alternatively, even a change point model would be preferable to the k-center approximation. Unlike the k-center approach, a change-point model accounts for temporal information, clustering observations that are temporally close – an essential aspect of drifting distributions.*
>
> The reviewer recommended rejection of the paper because we did not apply change point or change slope models. However, even a coarse reading of our paper would reveal that these models are irrelevant to our setting. The goal of our work is to deal with a drifting distribution. This includes a sequence of distributions that are always changing, with the magnitude of change varying arbitrarily at each step. The models suggested by the reviewer only deal with distributions that are stable or have constant change, until they change at change points. When a change point is discovered, the analysis drops all data before that point and analyzes the (assumed) stable (or constantly changing) distribution right after that point. Instead, the crux of our method is to decide over a window of data depending on the drift within that window, and it can take into account arbitrary drift patterns.
>
> The $(k,2)$-approximation  ($k$-means) as an objective to approximate a distribution is well-motivated and has been widely studied in the i.i.d. setting (e.g., Bartlett et al., 1998; Linder, 2000, 2002; Levrard, 2013, 2015; Biau et al., 2008; Maurer and Pontil, 2010; Maurer, 2016; Foster and Rakhlin, 2019; Li and Liu, 2021; Bucarelli et al., 2023). Differently from what is claimed by the reviewer, our algorithm clearly is data-driven and accounts for temporal information:  it rigorously determines, at each step, the best window of past samples to use to obtain a $(k,z)$-approximation of the current distribution (see theorems in Section 3) under arbitrary drift patterns. Change-point and slope models are generic models that are often used by practitioners to address particular drift patterns and are not specific to clustering. We do not see why the existence of those methods undermines the novelty of our theoretical results.
>
> **Reviewer's comment**: *Yet, in this study, which focuses on two-dimensional data with a scaled identity matrix as the covariance matrix, these challenges do not arise, making the changing slope model an ideal fit.*
>
> This is not true, since our method handles distributions over $\mathbb{R}^d$ with bounded support and arbitrary drift patterns (Theorem 2 in Section 3). The experimentation on mixtures of two-dimensional Gaussians is provided only as an explanatory instantiation of our general method.

---

### Author Rebuttal · Authors · 2024-11-23

We thank the reviewers for providing useful feedback on our paper. For each reviewer, we use the individual comment boxes to provide a separate comment addressing their concerns.

---

### Meta-Review · Area_Chair_fiPz · 2024-12-14

**Recommendation:** Accept
**Confidence:** 4

**Metareview:**

The paper studies k-clustering in the following model: at time t, there is a distribution $D_t$ from which _one sample point_ $x_t$ is given. The goal is to compete, at a given step T, with the best k-clustering for $D_T$. This is obviously impossible if $D_t$s change arbitrarily, so the authors use a parameter $\tau (w)$ that is the maximum distance (defined appropriately) between the distributions $D_T$ and $D_{T-i}$, for i within some "window" length $w$. Their algorithm obtains a bound that competes with the best-possible window length.

The paper also has lower bounds proving that some dependence of this form is necessary.

The paper is a good fit for ALT and I recommend accepting. The changing distributions model is obviously interesting, but there are some weaknesses that the authors should address/discuss in the final version: first, the model can be motivated a bit better -- what is a good application where distributions change slowly? why this specific choice of $\tau(W)$? Especially given that the lower bound has some slack, the latter question may be quite relevant.

Two of the reviewers point out the strengths, and the fact that the model is natural. The third reviewer (who is quite negative) raises questions about motivation that are definitely important, but I still think that given all the work the authors refer to on very similar models, this is not as critical. Moreover, the approaches suggested by the reviewer, while obviously relevant to the problem, do not appear to be the _only_ way to attack the problem.

**Paper Award:**

No